# Alignment-Aware Model Extraction Attacks on Large Language Models

## Abstract

Model extraction attacks (MEAs) on large language models (LLMs) have received increasing attention in recent research. However, existing attack methods typically adapt the extraction strategies originally developed for deep neural networks (DNNs). They neglect the underlying inconsistency between the training tasks of MEA and LLM alignment, leading to suboptimal attack performance. To tackle this issue, we propose Locality Reinforced Distillation (LoRD), a novel model extraction algorithm specifically designed for LLMs. In particular, LoRD employs a newly defined policy-gradient-style training task that utilizes the responses of victim model as the signal to guide the crafting of preference for the local model. Theoretical analyses demonstrate that I) The convergence procedure of LoRD in model extraction is consistent with the alignment procedure of LLMs, and II) LoRD can reduce query complexity while mitigating watermark protection through exploration-based stealing. Extensive experiments on domain-specific extractions validate the superiority of our method in extracting various state-of-the-art commercial LLMs. Our code is available at: https://anonymous.4open.science/r/LoRD-MEA-1EF2/.

## 1 Introduction

In recent years, we have witnessed the remarkable success of large language models (LLMs) such as ChatGPT (cha, 2024), Gemini (Anil et al., 2024), and Claude (cla, 2024), which are now widely employed in various consumer and industrial applications. Despite their success, these models may suffer from *model extraction attacks* (MEAs) (Krishna et al., 2020; Rafi et al., 2022; Xu et al., 2022; Li et al., 2023b), where their knowledge could be at risk of being stolen by an adversary through a *local model* that learns on the data collected from the *victim model*. Besides of some "open-source" LLMs (e.g., Alpaca (Taori et al., 2023)), which are trained on the chat history of GPT-4, cases of commercial model theft among companies have also been reported recently (Heath, 2023).

Under such a real-world threat, instead of focusing on MEAs against conventional DNNs, which have been extensively studied theoretically (Saad & Solla, 1995; Tian, 2020; Zhou et al., 2021) and empirically (Jagielski et al., 2020; Tramèr et al., 2016; Papernot et al., 2017),
a few recent works turn to explore model extrac-

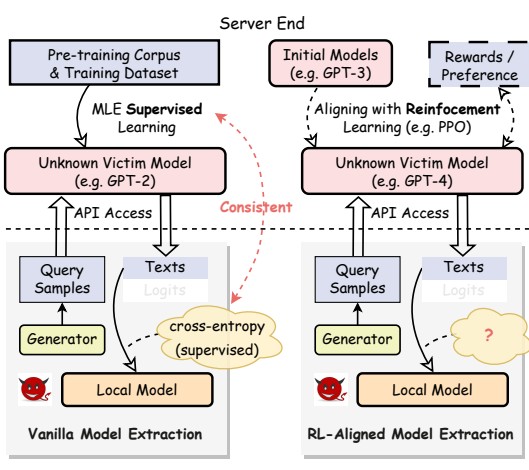

Figure 1: Comparison between vanilla MEAs on conventional DNNs (left) and MEAs on LLMs with alignments (right).

tion algorithms and theorems for LLMs. For example, Wallace et al. (2020) propose a monolingual-query-based imitation attack framework to steal machine translation knowledge from generative language models such as GPT-2. Li et al. (2023b) investigate threats of stealing the code-related knowledge from LLMs. However, these studies inherit those MEA algorithms from traditional fields,

such as computer vision (Tramèr et al., 2016; Papernot et al., 2017), and train the local model via supervised learning like maximum likelihood estimation (MLE) (Bengio et al., 2000; Myung, 2003), while neglecting the inconsistency of training tasks between MEAs and the alignments (Ouyang et al., 2022; Glaese et al., 2022; Bai et al., 2022a;b; Perez et al., 2023) of modern LLMs. As shown in Figure 1, modern LLMs typically employ alignments using reinforced learning, which is missing in the local model training of conventional MEAs. As a result, these attacks usually suffer from poor performance.

In this paper, we challenge the effectiveness of MLE in stealing a reinforcement-learning-aligned LLM, by analyzing its potential drawbacks as follows:

**Low query efficiency.** Current LLM-oriented MEAs suffer from unacceptably significant query times because they must collect enough generated responses, which entails exponential complexity in terms of generated tokens, resulting in low query efficiency.

**Vulnerability against defenses.** Directly learning from the responses of victim models can cause local models to inadvertently incorporate those *watermarks* (Cong et al., 2022; He et al., 2022; Zhao et al., 2022; He et al., 2021) embedded in the output of victim models. The residue of such watermarks makes the extraction process less stealthy and even serves as provenance evidence of model theft.

Motivated by these limitations, we propose Locality Rein-forced Distillation (LoRD), a query-efficient and watermark-resistant model extraction attack under a training paradigm with LLM's alignments. Stealing LLMs via reinforcement learning paradigms is challenging. The main reason is that the key component in the alignment procedure of LLMs, *reinforcement learning with human feedback* (RLHF) (Bai et al., 2022a;b; Perez et al., 2023), heavily relies on the feedback signal of **human annotators**, which is difficult to reproduce directly in the context of MEAs. To tackle this challenge, we develop a policy-gradient-style extraction procedure. This approach regards the *locality direction* between the generations of local models and victim models as the implicit reward signal. It can thus achieve a **human-feedback-free** reinforcement learning for our extraction attack. From the theoretical perspective, we show why those existing MEAs using *MLE* and *knowledge distillation (KD)* are inconsistent with the optimization procedure in LLMs' alignments. Along this way, we also demonstrate why LoRD can achieve stronger watermark resistance and higher query efficiency.

Extensive experiments on five downstream NLP tasks and ten datasets demonstrate that it is feasible to steal a commercial LLM with 175 billion parameters by a pre-trained local model with only 8 billion parameters under a given domain. The resulting local model performs statistically similar to the victim model for tasks not requiring extra knowledge (e.g., data-to-text), and only $0 \sim 3$ percentage lower for tasks requiring it (e.g., translation and QAs). This result poses an immediate threat of task-specific extraction on commercial LLMs. To further draw the capability boundary of such a threat, we also illustrate the "spectrum" in difficulties and upper bounds for extracting LLMs.

To summarize, the contributions of our paper are as follows:

**New Perspective of LLM Alignment for MEAs.** We present LoRD, a novel model extraction attack algorithm for LLMs. To our best knowledge, it is the first effective and realistic extraction algorithm that is compatible with the alignment procedure of LLMs.

**Theoretical Guarantee.** We theoretically prove that the convergence procedure of LoRD in MEAs is consistent with the alignments of LLMs. Furthermore, we demonstrate that LoRD can reduce query complexity while mitigating watermark protection through exploration-based stealing.

**Systematical Evaluation.** Extensive experiments on domain-specific extractions demonstrate that our method outperforms current extraction strategies across different downstream NLP tasks.

## 2 BACKGROUND

### 2.1 POLICY GRADIENT MODELS

Policy gradient models (PGM) are commonly used in reinforcement learning (RL) algorithms to optimize the agents based on the decided *action* of RL agents. Represented by TRPO (Schulman et al., 2015) and PPO (Schulman et al., 2017), policy gradient models minimize the the following

objective function:

$$\mathcal{L}_{pg,j} = -\hat{\mathbb{E}}_j[p_j^r(\theta)A_j], \tag{1}$$

where at each decision step $j$, $p_j^r(\theta) = \frac{\pi_\theta(a_j|s_j)}{\pi_{\theta_{old}}(a_j|s_j)}$ refers to the probability ratio defined by the optimized policy $\pi_\theta(a_j|s_j)$ and the initial policy $\pi_{\theta_{old}}(a_j|s_j)$, $s_j$ denotes the *state* of the environment, $a_j$ denotes the decided *action* of $\pi_\theta$, and $A_j$ is the *de-biased reward* of $a_j$. $A_j$ is estimated by the $Q$-value minus the $V$-value, i.e.,

$$A_j(s_j, a_j) = Q(s_j, a_j) - V(s_j). \tag{2}$$

Intuitively, $Q$-value refers to the *reward* if employing action $a_j$ at the given environment state $s_j$, which can be seen as the label of policy's decision. $V$-value represents the estimation of the expected reward at $s_j$. Consequently, $A_j$ denotes the *surprise* when taking action $a_j$.

To alleviate the *off the cliff* phenomenon that a large bad gradient update occurred from Equation 1, PGMs, such as PPO and TRPO, add some regularization terms to avoid large gradients. Specifically, TRPO constrains the distribution between $\pi_\theta$ and $\pi_{\theta_{old}}$ with KL divergence, and PPO warps a "clip" function to constrain the bounds of $p_j^r(\theta)$.

## 2.2 Language Modeling

**Supervised Training (SFT).** Given a pre-trained model with parameters $\theta$, supervised training is essentially the *maximum likelihood estimation (MLE)* task (Bengio et al., 2000; Myung, 2003), which fine-tunes $\theta$ on the labeled dataset $\mathcal{D}_{tr}^s = \{(\mathbf{x}_i, \mathbf{y}_i)|i = 1, 2, ..., N_{trs}\}$ by minimizing the following objective function:

$$\mathcal{L}_{mle} = -\prod_i^{N_{trs}} P_\theta(\mathbf{y}_i|\mathbf{x}_i) = -\prod_i^{N_{trs}} \prod_j^{N} P_\theta(y_{i,j}|\mathbf{x}_i, \mathbf{y}_{i,<j}), \tag{3}$$

where $N$ denotes the sequence length of $\mathbf{y}_i$, $y_{i,j}$ denotes the $j$-th token in $\mathbf{y}_i$, and $\mathbf{y}_{i,<j} = \{y_{i,0}, ..., y_{i,j-1}\}$. The logarithmic formula of Equation 3 can also be seen as a *joint cross-entropy* loss function:

$$\mathcal{L}_{ce} = -\sum_i^{N_{trs}} \log P_\theta(\mathbf{y}_i|\mathbf{x}_i) = -\sum_i^{N_{trs}} \sum_j^{N} \log P_\theta(y_{i,j}|\mathbf{x}_i, \mathbf{y}_{i,<j}). \tag{4}$$

Equation 4 is extensively utilized in LLM's pre-training and fine-tuning procedures. For instance, it can be applied to *instruction-following supervised fine-tuning (SFT)* with the training set $\mathcal{D}_{tr}$, wherein $\mathbf{x}_i$ encompasses the instruction and the task input, while $\mathbf{y}_i$ denotes the reference response.

Aligning LLMs merely by SFT is not always practical, as MLE tends to align the model with the one-hot distribution of $\mathbf{y}$, making it challenging to draw a sufficient variety of examples due to the *"exponential explosion"* of tokens (see Section 4 for more details). Moreover, providing standard answers for LLMs can sometimes be daunting for annotators, which further slows down and even degrades the alignment process through direct training.

Therefore, instead of "learning from answers" as in Equation 4, learning from *preferences* is proposed, which only requires the annotators to select a better response from a pair of texts generated by LLMs.

**Aligning from Preferences.** Employing reinforcement learning in LLMs typically consists of three stages. First, the annotators construct a preference dataset $\mathcal{D}^{pref} = \{(\mathbf{x}_i, \mathbf{y}_i^+, \mathbf{y}_i^-)\}$ by chatting with LLMs and rating their responses, where $\mathbf{y}_i^+$ and $\mathbf{y}_i^-$ denote the rated positive and negative responses of the dialogue context $\mathbf{x}_i$, respectively. Then, a *reward model* $R_{\theta_\phi}(\mathbf{x}, \mathbf{y}) \to \mathbf{r}$ is trained based on $\mathcal{D}^{pref}$ to simulate the environment and predict the reward values of tokens in given texts. It is trained with a pair-wise loss,

$$\mathcal{L}_r = -\sum_{(\mathbf{x}, \mathbf{y}^+, \mathbf{y}^-)\sim\mathcal{D}^{pref}} \sigma(R_{\theta_\phi}(\mathbf{x}, \mathbf{y}^+) - R_{\theta_\phi}(\mathbf{x}, \mathbf{y}^-)), \tag{5}$$

where $\sigma(\cdot)$ denotes the sigmoid function. Based on the reward model $R_{\theta_\phi}(\mathbf{x}, \mathbf{y})$, we can finally train the language models $P_\theta$ by maximizing its reward, i.e.,

$$\max_\theta \sum_{\mathbf{x}\sim\mathcal{D}_q} R_{\theta_\phi}(\mathbf{x}, \hat{\mathbf{y}}) - \beta\mathbb{D}_{KL}[P_\theta(\hat{\mathbf{y}}|\mathbf{x})||P_{\theta_{init}}(\hat{\mathbf{y}}|\mathbf{x})], \tag{6}$$

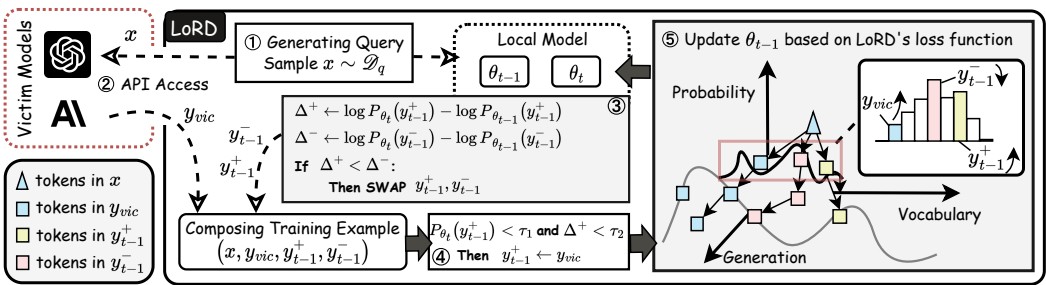

Figure 2: The stealing procedure of LoRD.

where $\mathcal{D}_q$ denotes the dataset of text inputs, $\hat{\mathbf{y}} \sim P_\theta(\mathbf{y}|\mathbf{x})$ denotes the sampled sequence of the training model, and $\theta_{init}$ is the initialized parameters of the model, e.g. the parameters after SFT. The Kullback-Leibler (KL) divergence term, $\beta \mathbb{D}_{KL}[P_\theta(\mathbf{y}|\mathbf{x})||P_{\theta_{init}}(\mathbf{y}|\mathbf{x})]$, introduced by TRPO (Schulman et al., 2015), is incorporated to constrain the shift of distribution in generated texts $\hat{\mathbf{y}}$, where $\beta$ is the hyperparameter.

Consequently, SFT shown in Equation 4 fine-tunes the pre-trained model with parameters $\theta_{pre}$ into an aligned model $\theta_{sft}$ through MLE, and RLHF outlined in Equation 6, further aligns $\theta_{sft}$ towards the target model $\theta_{vic}$. As this procedure is not consistent with the conventional training framework of DNNs, it remains unclear whether current MEAs (detailed in Appendix C.2) are effective and efficient in stealing a LLM. Specifically, we will first put forward a new stealing method in Section 3, and compare it with current MEAs in Section 4.

## 3 LoRD: LOCALITY REINFORCED DISTILLATION

### 3.1 OVERVIEW

In this subsection, we delve into the details of our model extraction framework, LoRD (Locality Reinforced Distillation). As described in Algorithm 1, LoRD follows a reinforcement learning paradigm, that is, it consists of several *periods*, and in each period, the model will learn to explore new responses and attempt to enhance the model trained in the last period. However, different from LLMs' alignments, the agent can neither obtain the reward from the reward model directly, nor label positive and negative responses manually. This motivates us to design a new RL method which can implicitly measure the reward for generated tokens under the guidance of victim model's responses.

Illustrated by Figure 2, LoRD first requires the model to sample two sentences randomly at period $t-1$, which are denoted as $\mathbf{y}_{t-1}^+$ and $\mathbf{y}_{t-1}^-$,

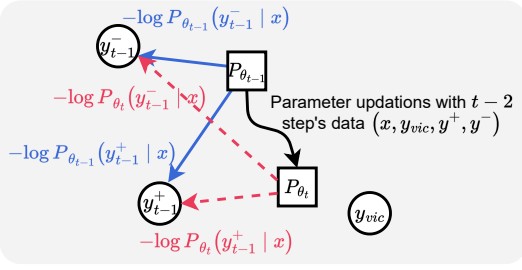

Figure 3: Determination of the positive and negative samples in LoRD. We sample $\mathbf{y}_{t-1}^+$ and $\mathbf{y}_{t-1}^-$ from $P_{\theta_{t-1}}(\cdot|\mathbf{x})$, and compute their conditional probabilities. The response with a higher probability increment on $\theta_t$ is selected as the positive sample.

respectively. In a new period $t$, it first computes the changes of likelihoods for these two sentences, among the old model $P_{\theta_{t-1}}$ and the current model $P_{\theta_t}$. These changes of likelihoods, denoted as $\Delta_t^+$ and $\Delta_t^-$, indicate whether a selected sentence is locally *isotropic* ($\Delta > 0$) to the optimization direction with victim model's response $\mathbf{y}_{vic}$ or not ($\Delta \leq 0$), which can be seen as the feedback signal for $P_{\theta_t}$ in the current optimization step. For convenience, we may swap $\mathbf{y}_{t-1}^+$ with $\mathbf{y}_{t-1}^-$ to make sure that $\Delta_t^+ > \Delta_t^-$ always holds. In this way, for pairs $(\mathbf{x}, \mathbf{y}_{vic})$ we can take $\mathbf{y}_{t-1}^+$ as a *locality neighborhood* of $\mathbf{y}_{vic}$ and $\mathbf{y}_{t-1}^-$ as the negative sample, all of which can be utilized in the training of $P_{\theta_t}$. Figure 3 illustrates this procedure. Additionally, LoRD takes $\mathbf{y}_{t-1}^+$ as the positive label under

the current scope only when $\Delta^+$ or $P_{\theta_t}(\mathbf{y}_{t-1}^+|\mathbf{x})$ exceed their respective fixed thresholds $\tau_1$ and $\tau_2$. If these conditions are not met, it will use $\mathbf{y}_{vic}$ as a substitute for $\mathbf{y}_{t-1}^+$ to enable a cold start.

Based on $\mathbf{y}_{vic}$, $\mathbf{y}_{t-1}^+$, and $\mathbf{y}_{t-1}^-$, we now design LoRD's loss function.

## 3.2 DESIGN OF LOSS FUNCTIONS

From Section 2.1, we know that the loss function of a policy gradient model can be expressed as an *objective function* to maximize the rewards of decisions (see Equation 1) and a *regularization term* to ensure the stability of training. Following this paradigm, the loss function of LoRD could be

$$\mathcal{L}_{\text{LoRD}} = \mathcal{L}_{obj} + \mathcal{L}_{reg}. \tag{7}$$

**Objective function $\mathcal{L}_{obj}$.** Inspired by the reward model $R_{\theta_\phi}$ existed in Equation 6, which is trained to distinguish between positive and negative samples, we propose utilizing the logarithmic proportion of positive to negative samples as the means of achieving a de-biased reward, i.e.,

$$\mathcal{L}_{obj} = -\sum_{\mathbf{x}\in\mathcal{D}_q} \log[\frac{P_{\theta_t}(\mathbf{y}_{t-1}^+|\mathbf{x})}{P_{\theta_t}(\mathbf{y}_{t-1}^-|\mathbf{x})}] = -\sum_{\mathbf{x}\in\mathcal{D}_q}[\log P_{\theta_t}(\mathbf{y}_{t-1}^+|\mathbf{x}) - \log P_{\theta_t}(\mathbf{y}_{t-1}^-|\mathbf{x})]. \tag{8}$$

Equation 8 exhibits similarities to previous studies on RL-enhanced LLM (Peters & Schaal, 2007; Peng et al., 2019; Go et al., 2023; Korbak et al., 2022; Rafailov et al., 2023). We provide a theoretical explanation for its consistency with the learning procedure of RLHF and the deduction procedure, as detailed in Section 4 and Appendix B.1.

However, training the local model merely by $\mathcal{L}_{obj}$ is ineffective due to two reasons: *i)* when $\mathcal{L}_{\text{LoRD}} := \mathcal{L}_{obj}$, no information from the victim model's responses is incorporated into the selection of $\mathbf{y}_{t-1}^+$ beyond the cold start phase, resulting in a meaningless *self-reward-based learning* loop for the stealing procedure; *ii)* the convergence of the local model's training cannot be guaranteed.

To address these two issues simultaneously, we design the regularization term as follows.

**Regularization loss $\mathcal{L}_{reg}$.** Different from LLM's RLHF (Schulman et al., 2015; Rafailov et al., 2023; Bai et al., 2022a) that typically constrain $\theta_t$ with initial model's generating distribution $P_{\theta_{init}}(\cdot|\mathbf{x})$ in RLHF, LoRD aims to directly constrain $\theta_t$ with victim model's distribution $P_{\theta_{vic}}(\cdot|\mathbf{x})$.

Unfortunately, $P_{\theta_{vic}}(\cdot|\mathbf{x})$ is typically **inaccessible** within the APIs of commercial LLMs and is not feasible for our black-box scenarios. Consequently, we incorporate the regularization techniques employed in PPO and TRPO but tailor our regularization as a bounded contrastive term between the likelihood of $\theta_t$ under the victim model's response and the negative sample, i.e.,

$$\mathcal{L}_{reg} = -\sum_{\mathbf{x}\in\mathcal{D}_q} clip(\log[\frac{P_{\theta_t}(\mathbf{y}_{vic}|\mathbf{x})}{P_{\theta_t}(\mathbf{y}_{t-1}^-|\mathbf{x})}]) = -\sum_{\mathbf{x}\in\mathcal{D}_q} clip(\log P_{\theta_t}(\mathbf{y}_{vic}|\mathbf{x}) - \log P_{\theta_t}(\mathbf{y}_{t-1}^-|\mathbf{x})). \tag{9}$$

In Equation 9, we utilize PPO's $clip(\cdot)$ function to limit the value of the regularization term, as we expect the regularization term could only be used to avoid the *off the cliff* problem (Schulman et al., 2017; 2015) in RL's convergence. Besides, our contrastive term can be seen as a streamlined black-box variant of the KL divergence in TRPO. This simplification offers two advantages: *i)* it alleviates the necessity of loading the initial model's weights, leading to a substantial reduction in GPU memory usage; *ii)* it eliminates the need for $P_{\theta_t}(\cdot|\mathbf{x})$, which would otherwise necessitate an additional exponential operation of $\log P_{\theta_t}(\cdot|\mathbf{x})$ that would slow down the forward computation process and increase extra consumption.[1]

Incorporating Equation 8 with Equation 9, we can reshape the loss function of LoRD as

$$\mathcal{L}_{\text{LoRD}} = \mathcal{L}_{obj} + \mathcal{L}_{reg} = \sum_{\mathbf{x}\in\mathcal{D}_q} \log[\frac{P_{\theta_t}(\mathbf{y}_{t-1}^-|\mathbf{x})}{P_{\theta_t}(\mathbf{y}_{t-1}^+|\mathbf{x})}] + clip(\log[\frac{P_{\theta_t}(\mathbf{y}_{t-1}^-|\mathbf{x})}{P_{\theta_t}(\mathbf{y}_{vic}|\mathbf{x})}])]. \tag{10}$$

Finally, we wrap $\mathcal{L}_{\text{LoRD}}$ with a sigmoid function $\sigma(\cdot)$ to normalize the loss to the interval $(0, 1)$, which is

$$\mathcal{L} = \sum_{\mathbf{x}\sim\mathcal{D}_q} \sigma(\log[\frac{P_{\theta_t}(\mathbf{y}_{t-1}^-|\mathbf{x})}{P_{\theta_t}(\mathbf{y}_{t-1}^+|\mathbf{x})}] + clip(\log[\frac{P_{\theta_t}(\mathbf{y}_{t-1}^-|\mathbf{x})}{P_{\theta_t}(\mathbf{y}_{vic}|\mathbf{x})}])). \tag{11}$$

---

[1] *logsoftmax* is preferred in the implementation of deep learning frameworks (tor), as the exponential operation in *softmax* and the logarithmic operation in *cross-entropy* can be canceled out by each other.

# 4 THEORETICAL ANALYSIS

This section will compare LoRD with current model extraction methods from a theoretical perspective. We will first reveal the underlying inconsistency between the optimization procedure of LLMs, which typically involves RL-based alignments, and the previous model extraction approaches utilizing *MLE* and *knowledge distillation (KD)*. Subsequently, we will demonstrate in theory the reasons why LoRD can achieve stronger watermark resistance and higher query efficiency than existing methods.

## 4.1 CONSISTENCY ANALYSIS REGARDING DIFFERENT LEARNING TASKS

Based on the analysis of the four objective functions for MLE, KD, RLHF and LoRD, we reach the proposition 1, and illustrate their convergence procedure exhibited in Figure 4. A detailed proof can be found in Appendix B.1.

**Proposition 1** (Consistency in Stealing Procedure). *The learning procedure for LLMs' alignments is consistent with the stealing procedure of LoRD, i.e., they both attempt to maximize the difference between the probabilities of positive and negative samples. Conversely, they are inconsistent with either MLE or KD. In MLE, the objective is maximizing the label probability, while KD aims to minimize the distance among all dimensions.*

Albeit the inconsistency in their *training procedures*, we put forward Proposition 2 to demonstrate that *with enough samples*, all these methods will reach the same distribution results.

**Proposition 2** (Equivalence when Converged). *Ideally, for any loss value of Equations 4, 5, 6, 10, or 11 converging to 0, we have* $\mathbf{y}^{+} \equiv \mathbf{y}_{vic}$. *Meanwhile, the local model's distribution* $P_{\theta}(\cdot|\mathbf{x})$ *will approach that of the victim model* $P_{\theta_{vic}}(\cdot|\mathbf{x})$ *on MEAs from all three discussed MEA methods, including LoRD, MLE, and KD.*

Proposition 2 ensures that the local model will converge to the victim model **regardless** of the choice of MEA methods. So what is the benefit of LoRD? In Section 4.2, we will show that LoRD outperforms current MEAs with two aspects: the query time reduction (i.e., size of the query set $\mathcal{D}_q$), and the watermark resistance of the learned local model.

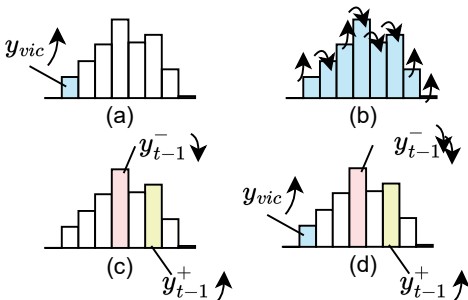

Figure 4: Illustrations for the converging procedure of probability distributions regarding four methods, namely MLE (a), KD (b), RLHF (c), and LoRD (d). Arrows indicate the expected optimization direction. We mark the distribution dimensions learned with labels in *blue*, and employ *pink* and *yellow* components to indicate the probabilities of positive and negative tokens, respectively.

## 4.2 COMPARATIVE ANALYSIS ON MODEL STEALING

**Query Efficiency.** Let $N_Q$ and $N_R$ denote the sequence lengths of the query text and the response text, respectively. For MLE, the *ideal* query numbers to populate the entire text space are given by $\mathcal{O}(V^{N_Q} \cdot V^{N_R})$, where $V$ represents the size of the vocabulary. In contrast, LoRD possesses the capability to automatically explore the generation token space, thereby significantly reducing the query requirements about generation candidates to a constant level. Specifically, the complexity of LoRD's query requirements is $\mathcal{O}(V^{N_Q} \cdot C)$, where $C$ is a constant that correlates with the capability of local models.

Based on the above analysis, a straightforward concern with employing MLE in LLMs' extraction is that, given the limited query times in real-world practices, it may suffer from incomplete learning, especially for text generation tasks. Consequently, the local model may tend to memorize some specific responses instead of achieving a broad understanding and generation. We call such a phenomenon *preference overfitting (PO)*, which indicates that the local model is only effective on a limited set of explored samples, and yet does not generalize well to unseen scenarios. In such cases, the local model usually exhibits a more "rugged" decision surface, which appears to *overfit* the preference sentences in $\mathcal{D}_{tr}$, as shown in Figure 11 (b). Figure 10 provides a visualization of it.

| | BLEU | | | | BERTScore | | | Rouge-L | | |
|---|---|---|---|---|---|---|---|---|---|---|
| | 1 | 2 | 3 | 4 | Pre. | Rec. | F1. | Pre. | Rec. | F1. |
| *Text to SQL: WikiSQL (Zhong et al., 2017) with 64 query samples* | | | | | | | | | | |
| Victim Model | 54.1 | 41.4 | 32.1 | 24.4 | 86.9 | 93.5 | 90.1 | 58.9 | 62.1 | 59.7 |
| Local Model | 20.2±0.2 | 14.5±0.2 | 10.9±0.1 | 8.1±0.1 | 82.5±0.0 | 92.4±0.1 | 87.1±0.0 | 22.6±0.3 | 66.4±0.4 | 33.2±0.3 |
| +MLE | 54.0±1.6 | 37.5±2.1 | 26.4±2.0 | 18.8±1.8 | 83.1±0.2 | 92.9±0.2 | 87.7±0.2 | 56.2±1.5 | 56.1±0.9 | 55.8±1.2 |
| +LoRD | 55.1±2.3 | 39.0±3.6 | 28.0±4.0 | 20.4±3.9 | 83.4±0.4 | 92.9±0.3 | 87.9±0.4 | 57.7±2.2 | 56.3±2.0 | 56.7±2.1 |
| *Text to SQL: Spider (Zhong et al., 2017) with 64 query samples* | | | | | | | | | | |
| Victim Model | 9.4 | 3.9 | 2.1 | 1.1 | 77.7 | 84.1 | 80.6 | 17.1 | 36.3 | 21.8 |
| Local Model | 6.4±0.2 | 2.1±0.1 | 0.9±0.1 | 0.5±0.0 | 80.0±0.1 | 82.6±0.1 | 81.2±0.1 | 10.0±0.3 | 21.5±0.6 | 12.7±0.4 |
| +MLE | 6.2±0.9 | 1.3±0.5 | 0.6±0.3 | 0.2±0.2 | 76.4±0.7 | 81.8±0.4 | 78.9±0.6 | 12.7±1.6 | 18.3±1.6 | 14.3±1.6 |
| +LoRD | 9.1±0.9 | 2.8±0.5 | 1.3±0.4 | 0.6±0.2 | 77.7±0.4 | 83.1±0.5 | 80.2±0.3 | 16.9±0.1 | 24.1±0.2 | 18.8±0.1 |
| *Data to Text: E2E NLG (Dušek et al., 2020) with 64 query samples* | | | | | | | | | | |
| Victim Model | 51.8 | 27.0 | 26.8 | 19.1 | 93.9 | 94.6 | 94.2 | 49.6 | 54.6 | 51.4 |
| Local Model | 31.1±0.1 | 20.1±0.2 | 13.5±0.2 | 8.9±0.3 | 86.1±0.1 | 92.4±0.1 | 89.1±0.1 | 29.0±0.3 | 49.4±0.4 | 35.9±0.3 |
| +MLE | 53.0±0.9 | 38.0±0.6 | 27.5±0.5 | 19.9±0.4 | 89.1±0.0 | 94.5±0.0 | 91.8±0.0 | 48.3±0.5 | 54.2±1.4 | 50.4±0.9 |
| +LoRD | 53.1±1.1 | 38.2±0.9 | 27.8±0.7 | 20.2±0.5 | 89.1±0.1 | 94.5±0.1 | 91.7±0.1 | 48.3±0.7 | 53.5±1.4 | 50.2±0.9 |
| *Data to Text: CommonGen (Lin et al., 2020) with 64 query samples* | | | | | | | | | | |
| Victim Model | 33.3 | 18.5 | 11.1 | 6.9 | 91.3 | 92.1 | 91.7 | 33.6 | 40.7 | 36.1 |
| Local Model | 12.2±0.0 | 6.5±0.1 | 3.8±0.0 | 2.3±0.0 | 83.0±0.0 | 89.7±0.0 | 86.2±0.0 | 14.6±0.1 | 46.2±0.2 | 21.6±0.0 |
| +MLE | 32.4±2.0 | 18.3±1.3 | 10.9±1.0 | 6.6±0.7 | 84.2±0.1 | 91.7±0.0 | 87.8±0.0 | 31.7±2.4 | 41.1±0.4 | 35.1±1.6 |
| +LoRD | 32.1±1.3 | 18.0±0.9 | 10.7±0.5 | 6.4±0.3 | 84.1±0.0 | 91.6±0.1 | 87.7±0.0 | 31.4±1.1 | 40.3±0.9 | 34.6±0.9 |
| *Summarization: TLDR (Kirk et al., 2023) with 64 query samples* | | | | | | | | | | |
| Victim Model | 11.9 | 5.0 | 2.6 | 1.5 | 85.9 | 88.4 | 87.1 | 13.4 | 30.9 | 18.4 |
| Local Model | 6.9±0.0 | 3.2±0.1 | 1.7±0.0 | 1.0±0.0 | 81.0±0.1 | 87.6±0.0 | 84.1±0.0 | 10.5±0.1 | 41.1±0.1 | 16.4±0.1 |
| +MLE | 10.6±0.5 | 4.8±0.2 | 2.6±0.1 | 1.6±1.1 | 83.6±0.7 | 88.4±0.2 | 85.9±0.5 | 14.3±0.5 | 32.7±1.1 | 18.9±0.4 |
| +LoRD | 10.2±0.3 | 4.5±0.1 | 2.4±0.1 | 1.4±0.0 | 84.1±0.1 | 88.3±0.1 | 86.2±0.1 | 12.8±0.3 | 33.2±0.9 | 18.0±0.2 |
| *Summarization: CNN Daily Mail (Hermann et al., 2015) with 64 query samples* | | | | | | | | | | |
| Victim Model | 20.4 | 10.8 | 6.4 | 4.1 | 86.4 | 87.8 | 87.1 | 22.4 | 40.8 | 28.2 |
| Local Model | 4.9±0.0 | 3.6±0.0 | 2.7±0.0 | 2.1±0.0 | 80.5±0.0 | 88.3±0.0 | 84.2±0.0 | 10.9±0.0 | 79.1±0.1 | 18.8±0.0 |
| +MLE | 5.1±0.5 | 3.7±0.0 | 2.8±0.0 | 2.2±0.0 | 80.6±0.0 | 88.3±0.0 | 84.3±0.0 | 11.3±0.1 | 78.6±0.1 | 19.3±0.1 |
| +LoRD | 5.3±0.0 | 3.9±0.0 | 2.9±0.0 | 2.3±0.0 | 80.6±0.0 | 88.4±0.0 | 84.3±0.0 | 11.3±0.1 | 78.6±0.2 | 19.1±0.1 |
| *Summarization: Samsum (Gliwa et al., 2019) with 64 query samples* | | | | | | | | | | |
| Victim Model | 20.7 | 11.4 | 6.9 | 4.4 | 88.1 | 91.7 | 89.8 | 24.2 | 50.5 | 31.6 |
| Local Model | 8.9±0.2 | 5.2±0.1 | 3.3±0.1 | 2.1±0.1 | 80.9±0.2 | 90.1±0.1 | 85.2±0.2 | 17.0±0.3 | 61.8±0.5 | 25.5±0.4 |
| +MLE | 16.9±1.1 | 9.4±0.7 | 5.8±0.4 | 3.7±0.3 | 83.9±0.9 | 90.9±0.6 | 87.3±0.8 | 25.2±0.8 | 49.8±2.5 | 31.0±1.7 |
| +LoRD | 18.4±0.7 | 10.1±0.3 | 6.0±0.2 | 3.7±0.1 | 84.9±0.1 | 91.5±0.1 | 88.1±0.1 | 23.2±0.8 | 49.7±1.5 | 30.2±0.6 |

Table 1: MEA comparison on three tasks, including structured text generation, data to text, and summarization. We use GPT-3.5-turbo as the victim model, and Llama3-8B (lla, 2024) as the local initial model. The *intensity* of the red or blue color corresponds to the degree of underperformance or outperformance relative to the victim model. More experiments are in Table 2 and Table 6.

**Watermark Resistance.** Another limitation of prevalent objective functions, such as MLE and KD, is their susceptibility to watermarks (Cong et al., 2022; He et al., 2022; 2021; Kirchenbauer et al., 2023) of output contents, i.e., while stealing knowledge from LLMs via responses $\mathbf{y}_{vic}$, watermarks within them will also been passively inherited by the local model. Consequently, the generated sentences of the local model may possess some *residual* of watermarks, which might be detected as evidence of stealing.

Despite introducing current watermark removal techniques, we indicate that LoRD can mitigate the influences of watermarks naturally, as it does not learn the likelihood of victim models' responses $\mathbf{y}_{vic} \sim \mathcal{D}_{tr}$ directly, but relies on $\mathbf{y}_{vic}$ to determine positive and negative labels from responses generated by the local model.

As depicted in Equation 8, LoRD guides the local model to learn the likelihood of $\mathbf{y}_{t-1}^{+}$ instead of $\mathbf{y}_{vic}$, which means that it will not been influenced by watermarks contained in $\mathbf{y}_{vic}$ explicitly. However, the regularization term $\mathcal{L}_{reg}$, as well as the replacement $\mathbf{y}_{t-1}^{+} \leftarrow \mathbf{y}_{vic}$ for a cold start, will indeed introduce watermarks from $\mathbf{y}_{vic}$. To address this, we can reshape Equation 11 into a convex combination of the objective function and the regularization, i.e.,

$$\mathcal{L} = \mathbb{E}[(1 - \lambda_1) \cdot (\log P_{\theta_t}(\mathbf{y}_{t-1}^{+}|\mathbf{x}) - \log P_{\theta_t}(\mathbf{y}_{t-1}^{-}|\mathbf{x})) + \lambda_1 \cdot clip(\log P_{\theta_t}(\mathbf{y}_{vic}|\mathbf{x}) - \log P_{\theta_t}(\mathbf{y}_{t-1}^{-}|\mathbf{x}))],$$

where $0 \leq \lambda_1 \leq 1$ is the hyperparameter.

When $\lambda_1$ is small, the convergence of LoRD will substantially focus on maximizing $P_{\theta_t}(\mathbf{y}_{t-1}^{+}|\mathbf{x})/P_{\theta_t}(\mathbf{y}_{t-1}^{-}|\mathbf{x})$, with which the local model will exhibit a strong watermark resistance ability. When $\lambda_1$ increases, LoRD will tend to rely more on the guidance of $\mathbf{y}_{vic}$, resulting in a higher risk of introducing watermarks. In the case of $\lambda_1 = 1$, the local model will converge to the victim model without any exploration and watermark resistance, which might suffer from the same level of defense by watermarks.

From a global perspective, $\mathcal{L}_{obj}$ represents the exploration and the locality learning ability of LoRD, which can mitigate the influences of watermarks. On the other hand, $\mathcal{L}_{reg}$ ensures the stability of the training procedure. Therefore, $\mathcal{L}$ characterizes a trade-off via $\lambda_1$ between the stability and the diversity during stealing, and Equation 11 can be seen as a special case of $\mathcal{L}$ with $\lambda_1 = 0.5$.

## 5 EXPERIMENTS

### 5.1 SETTINGS

**Datasets.** We evaluate MEAs on five mainstream natural language generation (NLG) tasks, including *machine translation*, *text summarization*, *question answering*, *structured text generation*, and *data-to-text*. We select ten representative datasets: WMT16 (Bojar et al., 2016), TLDR (Kirk et al., 2023), CNN Daily Mail (Hermann et al., 2015), Samsum (Gliwa et al., 2019), WikiSQL (Zhong et al., 2017), Spider (Yu et al., 2018), E2E-NLG (Dušek et al., 2020), CommonGen (Lin et al., 2020), PIQA (Bisk et al., 2020), and TruthfulQA (Lin et al., 2021) as benchmarks for our domain-specific evaluation. These datasets cover most of the downstream tasks in natural language generation. We compare not only the stealing efficacy of different MEA methods, but also the stealing difficulty across different downstream tasks. Table 5 lists all datasets and backbones used in the paper.

**Baselines.** As described in Section 2.2 and 4.1, we compare LoRD with two types of model extraction methods: maximum likelihood estimation (MLE) and knowledge distillation (KD). For MLE and LoRD, we conduct MEAs under pure **black-box attack settings** (see Appendix D for more details of the threat model). For KD, the predicted distributions are used specifically under grey-box settings.

**Metrics.** For text generation tasks, we evaluate extracted models with a semantic-level and two lexical-level metrics, BERTScore (Zhang et al., 2020), BLEU (Papineni et al., 2002), and Rouge-L (Lin, 2004), all of which are commonly used in the NLG evaluation. Regarding reasoning tasks (e.g., QA), we use Precision, Recall, Accuracy, and F1 score as their evaluation metrics.

**Implementation Details.** We use Llama3-8B as the local model to learn the outputs generated by victim models. We set sequence length varying 128 to 4096 depending on the selected tasks, and learning rate $3 \times 10^{-5}$. Our experiments run on $2 \times 80$GB Nvidia Tesla A100. We execute each training five times and record the mean values and standard variances in the following sections. For LoRD, we set $\tau_1$ and $\tau_2$ to 0.8 and -0.1, respectively. Besides, we set the period number $N_t$ to 512, and use $\lambda_1 = 0.5$ as the default formation of the loss function. The victim model's response, $\mathbf{y}_{vic}$, is generated by token sampling with a temperature of 1, the default setting for current LLM APIs. The local model also uses token sampling, but with a temperature of 0.8 and Top-P probability clipping (Holtzman et al., 2019) at 0.98. We use this setting to enhance the stability of generation in local models. Note that we have not incorporated *sampling strategies* with their corresponding hyperparameters into the design of LoRD. We believe that MEAs considering sampling strategies could inspire more powerful MEA methods, and we leave these improvements for future work.

### 5.2 STEALING DOMAIN-SPECIFIC KNOWLEDGE

We first select GPT-3.5-turbo, a checkpoint of ChatGPT, as the basic victim model. This is because its API provides *probabilities* of candidate words when generating responses. We employ Llama3-8B (lla, 2024), a small LLM with only a 4.5% fraction of parameters than the victim model as our initial local model. Though this LaViSH (**La**rge-**Vi**ctim-**S**mall-**H**eist) setting contradicts previous assumptions (Tramèr et al., 2016; Papernot et al., 2017; Jagielski et al., 2020) in MEA that the copy model should usually be "wider" or "larger" than the victim model to contain its knowledge, we believe this setting is more applicable in real world scenarios (Li et al., 2023b). Appendix D provides more detail for this setting. Besides, the number of query times selected in this section is less than 100, a significant degradation compared to previous studies (Li et al., 2023b). This is because, in our experiments, copy models can easily learn the knowledge with a few training samples and then exhibit only slight improvements afterward. More discussions on query times can be found in Appendix A.1.1.

**Fidelity and limits on stealing.** We first examine the fidelity and limits of a small LLM to steal commercial LLMs. As shown in Table 1, we list the performance of the victim model and the local model on three tasks, and provide two MEA methods, local model fine-tuned with MLE (+MLE) and LoRD (+LoRD), respectively. In Table 1, cells highlighted in *red* indicate poorer outcomes compared to the victim model, whereas *blue* signifies results that are on par or potentially superior to the victim model. The *intensity* of the red or blue color corresponds to the degree of underperformance or outperformance relative to the victim model.

We can see that the original performance of the local model is significantly lower than the victim model, i.e., with a 50% decrease in BLEU-4 or $10 \sim 25$ decrease in Rouge-L. Once we employ MEAs in the local model, its performance rapidly boosts to nearly the same as the victim model, with $0 \sim 40\%$ points of gaps in BERTScore. These gaps are negligible (e.g. $< 1\%$ in summarization) in some tasks, but remain eminent in other tasks such as reasoning, structured text generation, and machine translation. This phenomenon indicates that domain-specific model extractions can effectively learn domain-specific abilities from victim models but may perform poorly if downstream tasks require extra knowledge, such as machine translation and QA. We provide a stealing comparison among different local models in Table 9.

**Comparison among stealing methods.** Tables 1, 6, and 2 compare the stealing efficacy between MLE and our LoRD. The results consistently show that LoRD significantly outperforms MLE under the same MEA settings. Besides, for challenging tasks such as reasoning and translation, LoRD exhibits much higher improvements, which demonstrates that it can address the preference overfitting problem discussed in Section 4.2 and do enable the local model to learn the task ability from victim models. However, we also observe that for some tasks (e.g., summarization), LoRD shows no statistical difference from MLE, probably because these tasks are relatively simple, where merely MLE has already achieved comparable results to victim models.

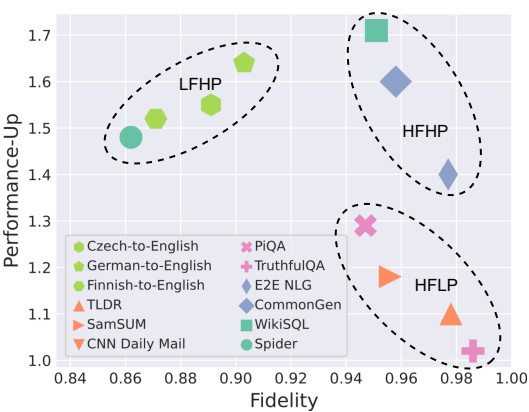

Figure 5: Spectrum of the fidelity and performance-up on extracting different downstream tasks. Current datasets can be divided into three groups: high fidelity and high performance-up (HFHP), high fidelity but low performance-up (HFLP), and low fidelity but high performance-up (LFHP).

**Tasks difficulties comparison.** Based on previous analysis, we observe that the performance and limitations of MEA depend on the category of tasks. Additionally, sometimes datasets in the same task exhibit significant differences in stealing. We put forward two metrics to measure task difficulties: the *fidelity* that measures extraction efficacy compared to victim models, and the *performance-up*, which assesses the performance gain before and after stealing for a given local model. Formally, given a test set $\mathcal{D}_{te} = \{(\mathbf{x}, \mathbf{y})\}$ and a corresponding metric $\mathcal{M}(hypothesis, reference)$, the fidelity ($F$) and performance-up ($P$) of the local model $\theta_{N_t}$ can be defined as:

$$F = \frac{\sum\limits_{\mathbf{x},\mathbf{y} \in \mathcal{D}_{te}} \mathcal{M}(\mathbf{y}_{N_t}, \mathbf{y})}{\sum\limits_{\mathbf{x},\mathbf{y} \in \mathcal{D}_{te}} \mathcal{M}(\mathbf{y}_{vic}, \mathbf{y})}, P = \frac{\sum\limits_{\mathbf{x},\mathbf{y} \in \mathcal{D}_{te}} \mathcal{M}(\mathbf{y}_{N_t}, \mathbf{y})}{\sum\limits_{\mathbf{x},\mathbf{y} \in \mathcal{D}_{te}} \mathcal{M}(\mathbf{y}_0, \mathbf{y})}, \quad (12)$$

where $\mathbf{y}_{N_t} \sim P_{\theta_{N_t}}(\cdot|\mathbf{x})$, $\mathbf{y}_0 \sim P_{\theta_0}(\cdot|\mathbf{x})$, and $\mathbf{y}_{vic} \sim P_{\theta_{vic}}(\cdot|\mathbf{x})$ denote the sampled responses from the trained local model ($\theta_{N_t}$), the initial local model ($\theta_0$), and the victim model ($\theta_{vic}$), respectively. In Figure 5, we illustrate a "spectrum" of extracting various downstream tasks based on these two metrics defined in Equation 12. The figure can assist in recognizing and defending commercial LLM's knowledge.

From Figure 5, we observe five tasks forming the following three scenario groups and datasets coming from the same tasks are mostly in the same group:

- High fidelity and high performance-up (HFHP). These tasks are challenging for a pre-trained model but can be effectively learned with the guidance of victim models. This group includes two tasks: data-to-text and structured text generation.

- High fidelity but low performance-up (HFLP). The initial local model already achieves a comparable performance to the victim model. QAs and summarization are in this group.

- Low fidelity but high performance-up (LFHP). While MEAs significantly improve the local model's performance, gaps between the local and victim models remain difficult to bridge with domain-specific extraction alone. Machine translation is a representative task whose reasons are explained in Section 5.2.

## 5.3 RESISTANCE TO WATERMARKS

Current LLM watermarking methods have been shown (Kirchenbauer et al., 2023) to be robust against commonly used erasing strategies (e.g., rephrasing), making watermark removal a distinct challenge. In this section, we validate the inherent resistance of LoRD to watermarks, suggesting that LoRD is preliminarily resistant to text watermarking. As described in Section 4, we highlight that LoRD can extract the victim models' knowledge with two terms: the straightforward likelihood learning term $\log P_{\theta_t}(\mathbf{y}_{vic}|\mathbf{x}) - \log P_{\theta_t}(\mathbf{y}_{t-1}^-|\mathbf{x})$ and the exploration term $\log P_{\theta_t}(\mathbf{y}_{t-1}^+|\mathbf{x}) - \log P_{\theta_t}(\mathbf{y}_{t-1}^-|\mathbf{x})$, where we can tune the hyperparameter $\lambda_1$ as shown in $\mathcal{L}$ to trade off the exploration and the convergence speed. Typically, a lower $\lambda_1$ encourages the model for conducting a slower but more diverse and localized exploration from its own generated text $\mathbf{y}_{t-1}^+$, potentially enhancing watermark resistance. In this subsection, we evaluate this analysis empirically.

**Watermarking Details.** Unlike previous experimental settings in Section 5, here we cannot utilize commercial LLMs as victim models due to the inability to control token sampling inside LLMs. Instead, we employ Llama3-70B as the victim model and watermark its outputs based on *"green" tokens selection*. Following prior research (Kirchenbauer et al., 2023), we separate the predicted vocabulary into a *green word set* and a *red word set* , assigning them randomly with the seed derived from the hash of generated tokens at the last generation step. Subsequently, we sample the next token **exclusively** from the green set, determined by a certain probability.

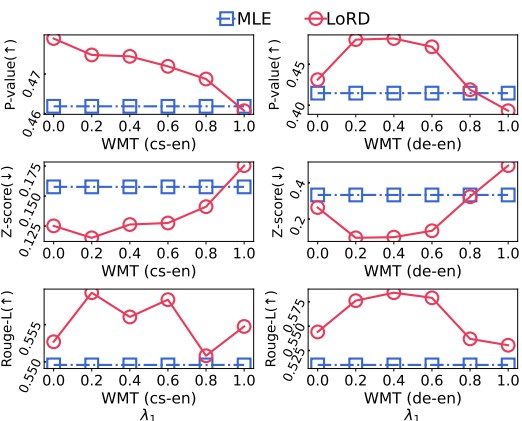

Figure 6: Comparison of watermarks resistance.

In this way, given the hypothesis $H_0$ that *texts are generated without the knowledge of the green word set*, we can estimate the probability $H_0$ occurs (*P-value*) and the *Z-score* of it for these texts. A high P-value, among with a low Z-score, indicates stronger watermark resistance for MEA algorithms.

**Result Analysis.** As depicted in Figure 6, we evaluate the watermark resistance for both MLE and LoRD, and demonstrate how LoRD's performance varies with different values of $\lambda_1$. The Z-score of LoRD witnesses a consistent increase as $\lambda 1$ arises, indicating that the "confidence" in rejecting the hypothesis, i.e., the risk to be suspected, arises when $\lambda_1$ increases. This finding coincides with the analysis in Section 4. However, $\lambda_1 = 0$ is a *abnormal* point in WMT (de-en), which might be because it disables the regularization term of LoRD's loss function. For tasks the local model does not own enough enough knowledge, it will lead to a significant performance degradation. Besides, we observe that the P-values of LoRD are generally higher than those of MLE when $\lambda_1$ is below 0.8, indicating that LoRD typically exhibits stronger watermarking resistance than MLE in most situations. It is noteworthy that this enhanced resistance seems not a "tax" of MEAs efficacy, as the Rouge-L (F1) scores of LoRD consistently surpass those of MLE and do not exhibit a significant negative correlation with their P-values.

## 6 CONCLUSION

In this paper, we have focused on the extraction problem of commercial large language models. We proposed LoRD, a practical and realistic extraction algorithm which is consistent with the alignment procedure of large language models. Our analysis proved that LoRD can reduce the query time significantly and mitigate the certification of current watermarks naturally, surpassing existing MEA algorithms' capabilities. Extensive experiments on domain-specific stealing demonstrated the superiority of our method.

## 7  ETHICAL CONSIDERATIONS

As discussed in Section 1, MEAs are becoming increasingly prevalent in industrial settings and have already been executed, yet there remains a critical gap in understanding which specific tasks are more susceptible and what capabilities are necessary for effective executions. This lack of knowledge exacerbates the challenges faced by LLM maintainers in safeguarding their systems. Our research can contribute to that. Besides, the theoretical problem we address (as shown in Section 4) offers a novel and insightful perspective on the nature of this threat. Based on these two points, **we believe the benefits of our paper outweigh potential harms, which aligns with the principles of the *Menlo Report* (Bailey et al., 2012) on ethics.** Additionally, we have submitted an anonymous version of the paper to the maintainers of the victim models used in our study to assist in improving their model security.

It is important to acknowledge, however, that the algorithms we propose could inadvertently enhance the efficiency of illicit extraction efforts by adversaries. To mitigate this risk, we have introduced and analyzed two defensive strategies in Appendix 8, assessing both their effectiveness and potential vulnerabilities under adaptive attack scenarios. This ensures a comprehensive approach to bolstering the security of LLMs.

## 8  POTENTIAL DEFENSES

**Query Detection.** One approach to effectively prevent the attack of LoRD is by detecting the distribution of query texts. This is because LoRD, similar to current MEA algorithms, makes no improvements to query samples, indicating that it can be detected by analyzing the statistical information of the adversary's queries, such as the number of queries, distribution of query contents, and so on. However, this defense is usually resource-consuming, as it requires the LLM provider to store all query texts of each user. Besides, the potential for false positives could adversely affect the user experience.

**More Powerful Watermarks.** While we highlight the watermark resistance of LoRD, watermarking remains one of the most effective solutions to mitigate MEAs. For example, some model-level watermarks, such as backdoor-based watermarking (Jia et al., 2021; Lv et al., 2024), can effectively certify the theft of DNNs. While model-level (e.g. backdoor-based) watermarks on pre-trained models raised increasing concerns recently (Peng et al., 2023a; Gu et al., 2022; Li et al., 2023a), model-level watermarking on LLMs remains preliminary. Besides, this technique might not work when the adversary only steals a subset of knowledge in which no backdoor is embedded.

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

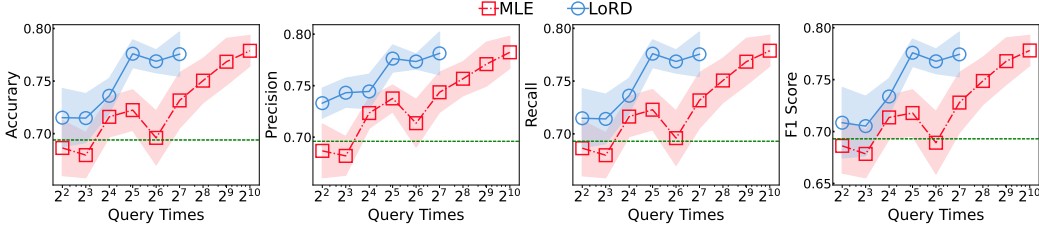

Figure 7: Comparison of query efficiency between MLE and LoRD on PiQA, where the *green horizontal line* represents the performance of the initialized local model. We increase query times for each method until reaching their bottlenecks. It can be found that the model extracted by LoRD typically performs a higher accuracy than MLE under the same number of queries. At the same time, LoRD reaches bottlenecks significantly earlier, reducing about 87% query cost compared with MLE.

# A  SUPPLEMENTAL EXPERIMENTS

## A.1  SCALING THE STEALING

In this subsection, we explore essential capacities to steal domain-specific knowledge from LLMs. We first analyze the influence of query times for the adversary, then compare the efficacy when utilizing different sizes of the local model, and finally compare the fidelity among different victim and local models.

### A.1.1  QUERY TIMES

We first investigate the influence of query numbers on MEAs. Specifically, we sample query examples randomly from the query dataset, starting from 4, and incrementally increase it until the performance of the learned model stabilizes. Figure 7 illustrates the stealing efficacy of LoRD and MLE on PiQA.

We observe that the scores of MLE and LoRD consistently increase as the query number rises, showing that a larger query number can improve

| Model/Metric | BLEU-1 | BLEU-4 | Rouge-L | BERTScore |
|---|---|---|---|---|
| *Czech to English with 16 query samples* | | | | |
| Victim Model | 0.611 | 0.313 | 0.604 | 0.957 |
| Local Model | 0.255 | 0.105 | 0.348 | 0.868 |
| +MLE | $0.535 \pm 0.01$ | $0.245 \pm 0.01$ | $0.526 \pm 0.01$ | $0.899 \pm 0.00$ |
| +LoRD | $0.545 \pm 0.01$ | $0.249 \pm 0.00$ | $0.538 \pm 0.01$ | $0.906 \pm 0.00$ |
| *German to English with 16 query sample* | | | | |
| Victim Model | 0.661 | 0.377 | 0.652 | 0.965 |
| Local Model | 0.276 | 0.130 | 0.359 | 0.877 |
| +MLE | $0.578 \pm 0.02$ | $0.302 \pm 0.01$ | $0.573 \pm 0.02$ | $0.904 \pm 0.01$ |
| +LoRD | $0.587 \pm 0.00$ | $0.308 \pm 0.00$ | $0.589 \pm 0.00$ | $0.917 \pm 0.00$ |
| *Finnish to English with 16 query samples* | | | | |
| Victim Model | 0.558 | 0.252 | 0.557 | 0.953 |
| Local Model | 0.242 | 0.085 | 0.320 | 0.866 |
| +MLE | $0.444 \pm 0.03$ | $0.173 \pm 0.02$ | $0.449 \pm 0.03$ | $0.905 \pm 0.00$ |
| +LoRD | $0.498 \pm 0.01$ | $0.196 \pm 0.00$ | $0.485 \pm 0.01$ | $0.905 \pm 0.00$ |

Table 2: MEA comparison on WMT16 (Bojar et al., 2016) among MLE and our LoRD methods, where we use GPT-3.5-turbo as the victim model, and Llama3-8B (lla, 2024) as the local initial model.

stealing efficacy steadily until reaching their empirical upper bounds. Additionally, LoRD typically obtains a higher score than MLE with the same number of queries, and reaches bottlenecks earlier, which can reduce the required query numbers by 87% compared to MLE. Moreover, in Figure 7, the performance of LoRD exhibits a relatively lower standard variance than MLE, indicating a more stable training procedure.

### A.1.2  SCALES OF LOCAL MODELS

As shown in our threat model (see Appendix D), we assume the adversary is stealing existing commercial LLMs with a small local model. This raises the question of selecting an appropriate interval of the local model's size. To address this concern, we illustrate the correlation between the local model's size and extraction efficacy on two machine translation tasks, Russian-to-English (ru-en) and German-to-English (de-en), as shown in Figure 8. Here, we employ seven OPT models (Zhang et al., 2022) as local models, with parameters ranging from 125 million to 30 billion, to minimize the interruptions of factors other than model size.

Figure 8 shows a sharp distinction between two machine translation tasks. In the de-en task, the performance of the local model increases steadily with model size, while this trend is not evident in the ru-en task with model size smaller than 30 billion. Nevertheless, the performance of a

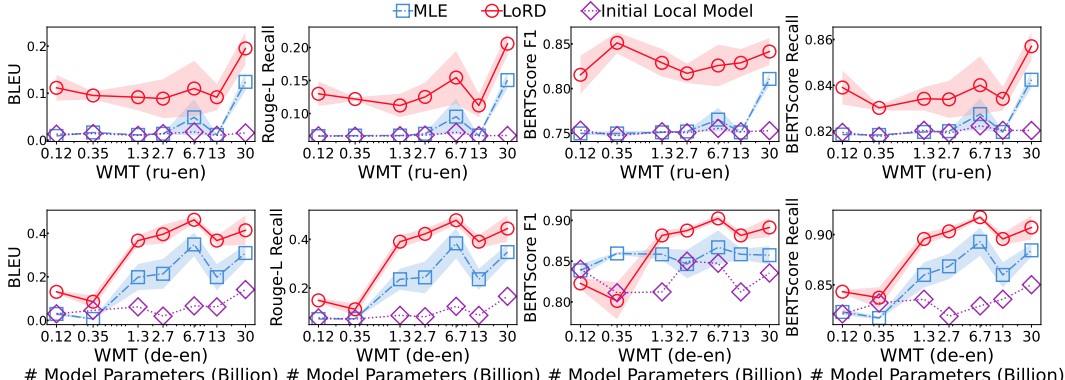

Figure 8: Experiments varying different model parameter scales.

30 billion parameter learned local model in ru-en cannot even be comparable to that of a 1.3 billion parameter local model in the de-en task. This phenomenon suggests that for tasks requiring commonsense knowledge, such as machine translation, the local model should at least possess foundational knowledge of the task (e.g., pre-trained on Russian texts) to learn from victim models effectively. Besides, experiments in BERTScore (F1) show that sometimes LoRD may underperform MLE when the local model has fewer than 1 billion parameters, demonstrating that it is challenging to bootstrap LoRD's exploration with a very small local model. By summarizing the increase in LoRD's curves, a model with 2.7 billion appears sufficient to steal domain-specific knowledge from commercial LLMs.

### A.1.3 FIDELITY UNDER DIFFERENT VICTIM AND LOCAL MODELS

We then evaluate the fidelity of extracting different victim models using various pre-trained local models. Specifically, we select GPT-3.5, GPT-4, and GPT-4o as victim models, and employ five state-of-the-art open-source models, Phi-3 (3.8B), OPT (6.7B), Qwen-2 (7B), Mistral-V3 (7B), and Llama-3 (8B), as local models, as shown in Figure 9.

Horizontally, while GPT-4 exhibits a consistently lower extracted fidelity compared to the other two victim models, vulnerabilities of the three victim models are generally similar. Vertically, fidelity of different local models can be significantly impacted by their performance. For instance, OPT (6.7B) shows a noticeably lower score compared to the other four models, which indicates that the initial performance of the local model will affect the performance of MEAs. Besides, Phi-3 (3.8B) achieves a comparable fi-

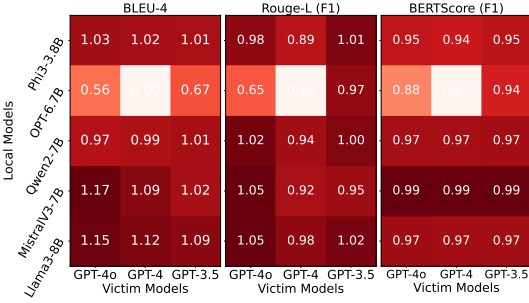

Figure 9: Fidelity of extracted models with different victim models (GPT-3.5-turbo, GPT-4, and GPT-4o) and different local models (Phi-3, OPT, Qwen2, MistralV3, and Llama3).

delity to larger models like Llama-3 (8B), demonstrating that the size of a local model does not influence final fidelity in domain-specific stealing after 2.7 billion, which corroborates the observation in Appendix A.1.2.

### A.2 VISUALIZATION OF DISTRIBUTIONS

We also investigate the *probability distributions* in the generation procedure among different extraction methods. Specifically, we visualize these distributions for four models, the victim model (GPT-3.5-turbo), the initial local model (llama3-8B), and the learned local models with MLE and LoRD.

| Model | DiaSafety | | | | | SafeRLHF | | | | |
|---|---|---|---|---|---|---|---|---|---|---|
| | Toxicity | Insult | Profanity | Severe Toxity | Threat | Toxicity | Insult | Profanity | Severe Toxity | Threat |
| Llama3-8B (initial) | 14..20 | 7.94 | 8.35 | 1.58 | 2.29 | 7.92 | 2.71 | 2.80 | 0.30 | 1.49 |
| +MLE | 8.31 | 3.69 | 4.31 | 0.83 | 1.50 | 4.87 | 1.98 | **1.66** | **0.16** | 1.02 |
| +LoRD | **6.45** | **2.81** | **3.56** | **0.71** | **1.34** | **3.55** | **1.15** | 2.84 | 0.38 | **0.79** |

Table 4: Comparison on safety alignment extraction tasks.

As plotted in Figure 10, each row in the subfigures refers to the distribution when generating the $i$-th token, with each column element indicating the *probability* predicted for the corresponding token index. We limit the visualization to no more than five token probabilities as currently only GPT-3.5-turbo provides the token prediction probabilities during generation, with a maximum of 5 candidate tokens (ope).

From Figure 10, we can see that both MLE and LoRD successfully redistribute the generation of the initial local model into a distribution similar to the victim model's, where probabilities, especially Top-1 tokens, have been well inherited in the extraction. This phenomenon supports our analysis in Proposition 2. However, distributions of MLE extracted models are consistently sharper than LoRD's, which aligns with our analysis in Section 4.2, where we claim that MLE leads local models to overfit to the preferred sentences (i.e., Top-1 tokens), namely *PO*, and thus to disrupt the original distributions, leveraging unusual low probabilities for other token indexes. The reason why LoRD can be resistant watermarks, i.e., tokens in Top-1, can also be derived from this discovery.

To compare MLE and LoRD accurately, we quantize the *entropy* of these distributions, and compute the *KL divergence* ($\mathbb{D}_{KL}$), and the

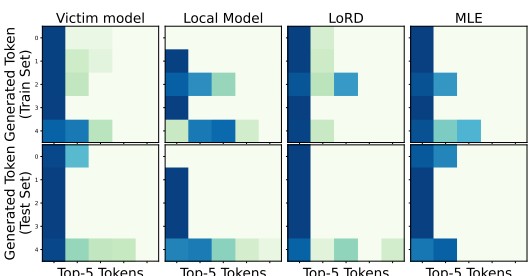

Figure 10: Token generation distributions of four models, namely the victim model, the (initial) local model, and the local model learned through LoRD and MLE, respectively. We visualize their logarithmic probability on examples sampled from the train set and test set, where a deeper color indicates a higher probability.

*Spearman Correlation (Spear. Corr.)* with respect to the victim and initial local model. As shown in Table 3, while the MLE extracted model exhibits a lower KL divergence (i.e., high distribution similarity) with the victim model than LoRD's on the training dataset, its KL divergence becomes comparable to LoRD's on the test set. Meanwhile, its Spearman correlation significantly decreases from 0.78 to 0.27, which shows that MLE cannot effectively imitate prediction behaviors of the victim model when encountering data beyond the training dataset.

A.3 STEALING SAFETY ALIGNMENTS

Besides of the domain-specific model extraction, we also propose the safety alignment extraction. Specifically, we select two popular safety alignment datasets for the experiments, namely SafeRLHF (Ji et al., 2024) and DiaSafety (Sun et al., 2022), to assess the safety of the generated responses. We employed PerspectiveAPI [2] to automatically evaluate the safety of the responses. We select five key aspects of safety probabilities: Toxicity, Insult, Profanity, Severe Toxicity, and Threat. In these categories, a lower score indicates better safety performance. For

| Models\Metrics | Entropy | To Victim Model | | To Initial Local Model | |
|---|---|---|---|---|---|
| | | $\mathbb{D}_{KL}\downarrow$ | Spear. Corr.↑ | $\mathbb{D}_{KL}$ | Spear. Corr. |
| *On training dataset* | | | | | |
| Initial Local Model | 0.395 | 0.503 | 0.620 | - | - |
| + LoRD | 0.209 | 0.051 | **0.880** | 0.169 | 0.680 |
| + MLE | 0.271 | **0.029** | 0.780 | 0.051 | 0.540 |
| *On the test dataset* | | | | | |
| Initial Local Model | 0.269 | 0.471 | 0.680 | - | - |
| + LoRD | 0.122 | 0.033 | **0.640** | 0.046 | 0.720 |
| +MLE | 0.275 | **0.032** | 0.274 | 0.001 | 0.740 |

Table 3: Quantization analysis on distributions. A low KL divergence or a high Spearman correlation indicates a high similarity.

the LoRD model, we have retained the same hyper-parameters as those used in our domain-specific experiments to ensure consistency.

---

[2]https://perspectiveapi.com/

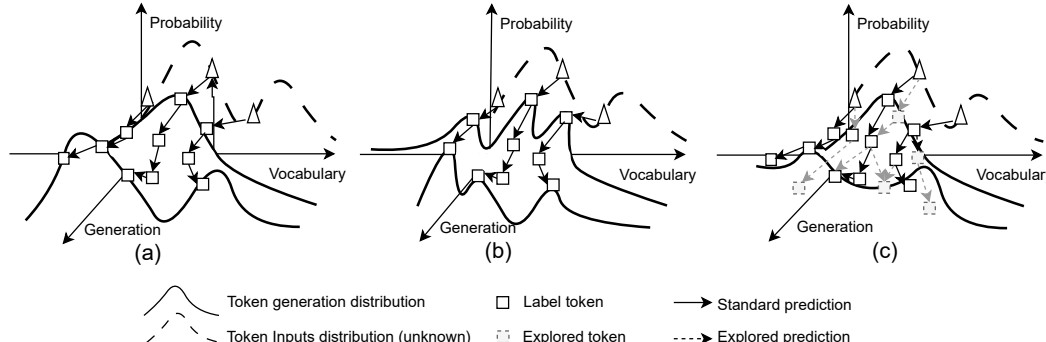

Figure 11: Comparison of learned joint *prediction distributions* among the victim model (a), local models are learned with MLE (b) and LoRD (c). Simply obtaining the tokens from the victim model (solid black squares), MLE may only memorize specific responses and build a complicated decision surface, resulting in *preference overfitting*. In contrast, LoRD further explores the candidate generation paths (dashed arrows and squares) under the guidance of the victim's generation, which is expected to better approximate the victim model in terms of generalization ability, especially under a limited query budget.

As shown in Table 4, we can see that both MLE and LoRD significantly reduce the harmful information after the stealing procedure. However, LoRD consistently outperforms MLE on most of the indicators, suggesting that it can achieve better performance in the alignment task.

## B PROOFS

### B.1 PROOFS OF PROPOSITION 1

As we described in Section 2, both existing methods and LoRD are learned from the victim model's response $\mathbf{y}_{vic}$ and the corresponding probability distribution $P_{\theta_{vic}}(\cdot|\mathbf{x}) \in \mathbb{R}^V$, where $V$ denotes the vocabulary size. Therefore, we first investigate how the local model is learned to emulate the distribution of the victim model, $P_{\theta_{vic}}(\cdot|\mathbf{x})$, under the following three stealing strategies.

**Expected Distribution of MLE.** We can first reshape the MLE loss into a special formation of Kullback-Leibler divergence with labels of one-hot distributions, that is,

$$\mathcal{L}_{ce} = - \sum_{\mathbf{x},\mathbf{y} \sim \mathcal{D}_{tr}} \log P_\theta(\mathbf{y}_{vic}|\mathbf{x}) = \sum_{\mathbf{x},\mathbf{y} \sim \mathcal{D}_{tr}} \sum_j^N \mathbb{D}_{KL}[\mathbf{1}_{y_{vic,j}} || P_\theta(\cdot|\mathbf{x}, \mathbf{y}_{vic,<j})], \quad (13)$$

where $\mathbf{1}_{y_{vic,j}}$ is a one-hot vector in which only $\mathbf{1}_{y_{vic,j}}[y_{vic,j}] = 1$ and all the other elements are 0. Equation 13 demonstrates that MLE learns to maximize the probability of $\mathbf{y}_{vic,j}$, without explicit constraints on probabilities across other dimensions.

**Expected Distribution of KD.** Following a previous work (Hinton et al., 2015), the objective function of KD is

$$\mathcal{L}_{kd} = \mathbb{D}_{KL}[P_{\theta_{vic}}(\cdot|\mathbf{x}) || P_\theta(\cdot|\mathbf{x})] + T^2 \cdot \mathbb{D}_{KL}[\text{SM}(P_{\theta_{vic}}(\cdot|\mathbf{x})/T) || \text{SM}(P_\theta(\cdot|\mathbf{x})/T)], \quad (14)$$

where $\text{SM}(\cdot)$ represents the *softmax function*, and $T > 1$ denotes the temperature to smooth the targeted distribution $P_{\theta_{vic}}(\cdot|\mathbf{x})$. As described in Equation 14, knowledge distillation aims to align $P_\theta(\cdot|\mathbf{x})$ with $P_{\theta_{vic}}(\cdot|\mathbf{x})$ in both the original and the smoothed probability across all dimensions, which is exceptionally comprehensive among these methods.

**Expected Distribution of Alignments.** Replacing Equation 6 with Equation 5, we can merge the optimization target of LLMs' alignments as

$$\min_{\theta*} - \sum_{(\mathbf{x},\mathbf{y}^+,\mathbf{y}^-) \sim \mathcal{D}^{pref}} \sigma\left( \log \frac{P_{\theta*}(\mathbf{y}^+|\mathbf{x})/P_{\theta*}(\mathbf{y}^-|\mathbf{x})}{P_{\theta_{init}}(\mathbf{y}^+|\mathbf{x})/P_{\theta_{init}}(\mathbf{y}^-|\mathbf{x})} \right)$$
$$\Rightarrow \max_{\theta*} \sum_{(\mathbf{x},\mathbf{y}^+,\mathbf{y}^-) \sim \mathcal{D}^{pref}} \log P_{\theta*}(\mathbf{y}^+|\mathbf{x}) - \log P_{\theta*}(\mathbf{y}^-|\mathbf{x}), \quad (15)$$

where $\theta*$ denotes the expected parameters of the models as

$$P_{\theta*}(\mathbf{y}|\mathbf{x}) = \frac{1}{Z(x)} P_{\theta_{init}}(\mathbf{y}|\mathbf{x}) \cdot e^{\frac{1}{\beta} R_\phi(\mathbf{x}, \mathbf{y})}. \quad (16)$$

We provide a detailed derivation for Equation 16 in Appendix B.2. By replacing Equation 15 with Equation 16, the expected distribution can be represented as $\mathbf{r}_{i,j} \cdot P_{\theta_{init}}(\cdot|\mathbf{x})$, in which $\mathbf{r}_{i,j}$ indicates the wrapped distribution gain. This distortion aims to maximize the ratio $P_\theta(y_j^+|\mathbf{x}, \mathbf{y}_{<j}^+)/P_\theta(y_j^-|\mathbf{x}, \mathbf{y}_{<j}^-)$, and leave the probabilities in other dimensions unconstrained directly.

**Expected Distribution of LoRD.** Similar to alignments, the expected converging procedure by the objective function $\mathcal{L}_{obj}$ is also intended to maximize the ratio between positive samples and negative samples, i.e., $P_{\theta_t}(\mathbf{y}_{t-1}^+|\mathbf{x})/P_{\theta_t}(\mathbf{y}_{t-1}^-|\mathbf{x})$. Meanwhile, the regularization term $P_{\theta_t}(\mathbf{y}_{vic}|\mathbf{x})/P_{\theta_t}(\mathbf{y}_{t-1}^-|\mathbf{x})$ will guide the models to maximize the ratio between $\mathbf{y}_{vic}$ and $\mathbf{y}_{t-1}^-$. As the "standard response" to be learned, $\mathbf{y}_{vic}$ can be viewed sufficiently as a positive example. Therefore, we can derive that the optimization target of LoRD is consistent with RLHF's optimization, i.e., both encourage local models to maximize the probability proportion between positive and negative samples.

Similar to Equation 16 in which the optimized model can be seen as the distortion of the original model $P_{\theta_{init}}$, in LoRD the optimized model can be regarded as the distortion of the local model $P_{\theta_0}$, with $P_{\theta_t}(\cdot|\mathbf{x}) = \mathbf{r}_{i,j}^t P_{\theta_{t-1}}(\cdot|\mathbf{x})$ at each step $t$, where the distortion term $\mathbf{r}_{i,j}^t$ is intended to jointly maximize $P_{\theta_t}(\mathbf{y}_{t-1}^+|\mathbf{x})/P_{\theta_t}(\mathbf{y}_{t-1}^-|\mathbf{x})$ and $P_{\theta_t}(\mathbf{y}_{vic}|\mathbf{x})/P_{\theta_t}(\mathbf{y}_{t-1}^-|\mathbf{x})$, while leaving the probabilities in other dimensions unconstrained directly.

## B.2 THE DEDUCTION OF EQUATION 16 IN PROPOSITION 1

From Equation 6, we can get that

$$\max_\theta \sum_{\mathbf{x} \sim \mathcal{D}_q} R_{\theta_\phi}(\mathbf{x}, \hat{\mathbf{y}}) - \beta \mathbb{D}_{KL}[P_\theta(\mathbf{y}|\mathbf{x})||P_{\theta_{init}}(\mathbf{y}|\mathbf{x})]$$

$$\Rightarrow \max_\theta \sum_{\mathbf{x} \sim \mathcal{D}_q} \sum_{\mathbf{y} \sim P_\theta(\cdot|\mathbf{x})} R_{\theta_\phi}(\mathbf{x}, \mathbf{y}) - \beta[\log P_\theta(\mathbf{y}|\mathbf{x}) - \log P_{\theta_{init}}(\mathbf{y}|\mathbf{x})]$$

$$\Rightarrow \min_\theta \sum_{\mathbf{x} \sim \mathcal{D}_q} \sum_{\mathbf{y} \sim P_\theta(\cdot|\mathbf{x})} -\frac{1}{\beta} R_{\theta_\phi}(\mathbf{x}, \mathbf{y}) + \log \frac{P_\theta(\mathbf{y}|\mathbf{x})}{P_{\theta_{init}}(\mathbf{y}|\mathbf{x})}$$

$$\Rightarrow \min_\theta \sum_{\mathbf{x} \sim \mathcal{D}_q} \sum_{\mathbf{y} \sim P_\theta(\cdot|\mathbf{x})} -\log(\exp(\frac{1}{\beta} R_{\theta_\phi}(\mathbf{x}, \mathbf{y}))) + \log \frac{P_\theta(\mathbf{y}|\mathbf{x})}{P_{\theta_{init}}(\mathbf{y}|\mathbf{x})}$$

$$\Rightarrow \min_\theta \sum_{\mathbf{x} \sim \mathcal{D}_q} \sum_{\mathbf{y} \sim P_\theta(\cdot|\mathbf{x})} \log \frac{P_\theta(\mathbf{y}|\mathbf{x})}{\exp(\frac{1}{\beta} R_{\theta_\phi}(\mathbf{x}, \mathbf{y})) \cdot P_{\theta_{init}}(\mathbf{y}|\mathbf{x})}.$$

If we define a partition function $Z(\mathbf{x})$ with the formation of

$$Z(\mathbf{x}) = \sum_{\mathbf{y}} P_{init}(\mathbf{y}|\mathbf{x}) \exp(\frac{1}{\beta} R_{\theta_\phi}(\mathbf{x}, \mathbf{y})), \quad (17)$$

we can reformat the optimization target as

$$\min_\theta \sum_{\mathbf{x} \sim \mathcal{D}_q} \sum_{\mathbf{y} \sim P_\theta(\cdot|\mathbf{x})} \log \frac{P_\theta(\mathbf{y}|\mathbf{x})}{\exp(\frac{1}{\beta} R_{\theta_\phi}(\mathbf{x}, \mathbf{y})) \cdot P_{\theta_{init}}(\mathbf{y}|\mathbf{x})}$$

$$\Rightarrow \min_\theta \sum_{\mathbf{x} \sim \mathcal{D}_q} \sum_{\mathbf{y} \sim P_\theta(\cdot|\mathbf{x})} \log \frac{Z(\mathbf{x}) \cdot P_\theta(\mathbf{y}|\mathbf{x})}{\exp(\frac{1}{\beta} R_{\theta_\phi}(\mathbf{x}, \mathbf{y})) \cdot P_{\theta_{init}}(\mathbf{y}|\mathbf{x})}$$

$$- \log Z(\mathbf{x}).$$

If we mark $\frac{1}{Z(\mathbf{x})} \exp(\frac{1}{\beta} R_{\theta_\phi}(\mathbf{x}, \mathbf{y})) \cdot P_{\theta_{init}}(\mathbf{y}|\mathbf{x})$ as $P_{\theta*}(\mathbf{y}|\mathbf{x})$, then we have

$$\min_{\theta} \sum_{\mathbf{x} \sim \mathcal{D}_q} \sum_{\mathbf{y} \sim P_{\theta}(\cdot|\mathbf{x})} \log \frac{Z(\mathbf{x}) \cdot P_{\theta}(\mathbf{y}|\mathbf{x})}{\exp(\frac{1}{\beta} R_{\theta_{\phi}}(\mathbf{x}, \mathbf{y})) \cdot P_{\theta_{init}}(\mathbf{y}|\mathbf{x})} - \log Z(\mathbf{x})$$

$$\Rightarrow \min_{\theta} \sum_{\mathbf{x} \sim \mathcal{D}_q} \sum_{\mathbf{y} \sim P_{\theta}(\cdot|\mathbf{x})} \log \frac{P_{\theta}(\mathbf{y}|\mathbf{x})}{P_{\theta*}(\mathbf{y}|\mathbf{x})} - \log Z(\mathbf{x}).$$

Because $Z(\mathbf{x})$ is independent to $\mathbf{y}$, we can deduct that

$$\min_{\theta} \sum_{\mathbf{x} \sim \mathcal{D}_q} \sum_{\mathbf{y} \sim P_{\theta}(\cdot|\mathbf{x})} \log \frac{P_{\theta}(\mathbf{y}|\mathbf{x})}{P_{\theta*}(\mathbf{y}|\mathbf{x})} - \log Z(\mathbf{x})$$

$$\Rightarrow \min_{\theta} \sum_{\mathbf{x} \sim \mathcal{D}_q} \left[ \sum_{\mathbf{y} \sim P_{\theta}(\cdot|\mathbf{x})} \log \frac{P_{\theta}(\mathbf{y}|\mathbf{x})}{P_{\theta*}(\mathbf{y}|\mathbf{x})} \right] - \log Z(\mathbf{x}) \tag{18}$$

$$\Rightarrow \min_{\theta} \sum_{\mathbf{x} \sim \mathcal{D}_q} \mathbb{D}_{KL}[P_{\theta}(\mathbf{y}|\mathbf{x}) || P_{\theta*}(\mathbf{y}|\mathbf{x})] - \log Z(\mathbf{x}).$$

As we know that $Z(\mathbf{x})$ does not contain $\theta$, the above optimization target actually minimizes the KL-divergence between the distribution of $P_{\theta}$ and $P_{\theta*}$, demonstrating that $\theta*$ is the optimal value of $\theta$ that satisfies

$$P_{\theta*}(\mathbf{y}|\mathbf{x}) = \frac{1}{Z(\mathbf{x})} \exp(\frac{1}{\beta} R_{\theta_{\phi}}(\mathbf{x}, \mathbf{y})) \cdot P_{\theta_{init}}(\mathbf{y}|\mathbf{x}). \tag{19}$$

Based on equation 19, we can see that the optimal distribution of $\theta$ is built upon $P_{\theta_{init}}$ with a distortion, as we discussed in Section 4.1.

### B.3 PROOFS OF PROPOSITION 2

**Guarantee of MLE.** From Equation 13 we can obtain that when $\mathcal{L}_{ce}$ decreases to 0, the KL divergence between $P_{\theta}(\cdot|\mathbf{x})$ and $P_{\theta_{vic}}(\cdot|\mathbf{x})$ decreases to 0, indicating that $P_{\theta}(\cdot|\mathbf{x})$ equals to $P_{\theta_{vic}}(\cdot|\mathbf{x})$.

**Guarantee of KD.** As we know, $\mathbb{D}_{KL}(p, q) \geq 0 \ \forall \ p$ and $q$. Therefore, if $\mathcal{L}_{kd}$ shown in Equation 14 equals to 0, then both $\mathbb{D}_{KL}[P_{\theta}(\cdot|\mathbf{x}) || P_{\theta_{vic}}(\cdot|\mathbf{x})]$ and $\mathbb{D}_{KL}[\text{SM}(P_{\theta}(\cdot|\mathbf{x})/T) || \text{SM}(P_{\theta_{vic}}(\cdot|\mathbf{x})/T)]$ equal to 0. For the latter one, we have

$$\mathbb{D}_{KL}[\text{SM}(P_{\theta_{vic}}(\cdot|\mathbf{x})/T) || \text{SM}(P_{\theta}(\cdot|\mathbf{x})/T)]$$

$$= \mathbb{E}_{\mathbf{y} \sim P_{\theta_{vic}}(\cdot|\mathbf{x})} \mathbb{E}_{y \in \mathbf{y}} \left[ \log \frac{\exp(P_{\theta}(y|\mathbf{x}, \mathbf{y}_p)/T) / \sum_{y' \in \mathbf{y}} \exp(P_{\theta_{vic}}(y|\mathbf{x}, \mathbf{y}_p)/T)}{\exp(P_{\theta_{vic}}(y|\mathbf{x}, \mathbf{y}_p)/T) / \sum_{y' \in \mathbf{y}} \exp(P_{\theta}(y|\mathbf{x}, \mathbf{y}_p)/T)} \right]$$

$$= \mathbb{E}_{\mathbf{y} \sim P_{\theta_{vic}}(\cdot|\mathbf{x})} \mathbb{E}_{y \in \mathbf{y}} \left[ \log \frac{\exp((P_{\theta}(y|\mathbf{x}, \mathbf{y}_p) - P_{\theta_{vic}}(y|\mathbf{x}, \mathbf{y}_p))/T)}{\sum_{y' \in \mathbf{y}} \exp(P_{\theta_{vic}}(y|\mathbf{x}, \mathbf{y}_p)/T) / \sum_{y' \in \mathbf{y}} \exp(P_{\theta}(y|\mathbf{x}, \mathbf{y}_p)/T)} \right],$$

where we can observe that only when $P_{\theta}(\cdot|\mathbf{x})$ equals to $P_{\theta_{vic}}(\cdot|\mathbf{x})$ can this term reduce to 0. Integrating the analysis of these two terms, we can obtain that $\mathcal{L}_{kd} = 0$ represents the local model's distribution converge to that of the victim model.

**Guarantee of LoRD.** When $\mathcal{L}$ shown in Equation 11 equals to 0, the proportion of $P_{\theta_t}(\mathbf{y}_{vic}|\mathbf{x})/P_{\theta_t}(\mathbf{y}_{t-1}^-|\mathbf{x})$ and $P_{\theta_t}(\mathbf{y}_{t-1}^+|\mathbf{x})/P_{\theta_t}(\mathbf{y}_{t-1}^-|\mathbf{x})$ should limit to $-\infty$. As we know that *i)* in a distribution $\sum P_{\theta_t}(\cdot|\mathbf{x}) = 1$ and *ii)* $\mathbf{y}_{t-1}^+$ is a dynamic positive response generated at each period, we can deduct that when $\mathcal{L} = 0$ there must be $\mathbf{y}_{vic} = \mathbf{y}_{t-1}^+$, i.e., $P_{\theta_t}(\mathbf{y}_{vic}|\mathbf{x}) = P_{\theta_t}(\mathbf{y}_{t-1}^+|\mathbf{x}) = 1$ and $P_{\theta_t}(\mathbf{y}_{t-1}^-|\mathbf{x}) = 0$. Note that this is merely a theoretical limit that cannot be reached, because $\mathbf{y}_{t-1}^-$ will not be sampled if its probability is 0, and $\mathbf{y}_{t-1}^+$ usually doesn't exhibit a significant distinction to $\mathbf{y}_{t-1}^-$ when sampling.

| Datasets\ Models | Links |
|---|---|
| PIQA | https://huggingface.co/datasets/piqa |
| TruthfulQA | https://huggingface.co/datasets/truthful_qa |
| WMT16 | https://huggingface.co/datasets/wmt16 |
| E2E NLG | https://huggingface.co/datasets/e2e_nlg |
| CommonGen | https://huggingface.co/datasets/allenai/common_gen |
| WikiSQL | https://huggingface.co/datasets/wikisql |
| Spider | https://huggingface.co/datasets/spider |
| TLDR | https://huggingface.co/datasets/UCL-DARK/openai-tldr-filtered |
| SamSUM | https://huggingface.co/datasets/samsum |
| CNN Daily Mail | https://huggingface.co/datasets/cnn_dailymail |
| Llama3-8B | https://huggingface.co/meta-llama/Meta-Llama-3-8B-Instruct |
| Llama3-70B | https://huggingface.co/meta-llama/Meta-Llama-3-70B-Instruct |
| Phi3-3.8B | https://huggingface.co/microsoft/Phi-3-mini-4k-instruct |
| OPT-6.7B | https://huggingface.co/facebook/opt-6.7b |
| Qwen2-7B | https://huggingface.co/Qwen/Qwen2-7B-Instruct |
| MistralV3-7B | https://huggingface.co/mistralai/Mistral-7B-Instruct-v0.3 |

Table 5: Datasets and pre-trained model checkpoints used in the paper.

## C  SUPPLEMENTAL RELATED WORKS

### C.1  HUMAN-FEEDBACK-FREE ALIGNMENTS

There are several alternatives to the standard RLHF approach. Lee et al. (2023) propose reinforcement learning with AI feedback (RLAIF) as a means to diminish the annotation burden associated with the preference assessments. Besides, there are some approaches, such as direct preference optimization (DPO) (Rafailov et al., 2023), that conceptualize the language model itself as the reward model and thus consolidate Equation 5 and Equation 6 into a unified supervised and preference-based training task. Since they do not change the primary targets (i.e., maximizing rewards) and optimization strategies of LLM's alignments, we only consider the standard formation of alignments for simplicity in our theoretical analysis.

### C.2  LANGUAGE MODELS EXTRACTION

Studies to steal language models originated from the natural language understanding (NLU) models, such as BERT(Devlin et al., 2019), and then evolved to generative language models, especially large language models recently.

Krishna et al. (2020) highlights early recognition of model extraction threats in language models. By constructing text inputs with randomly vocabulary sampling, they successfully extract the weights from BERT-based APIs. Besides, Rafi et al. (2022) investigate the feasibility of side-channel model extraction attacks, revealing that by analyzing extra signals from GPU kernels, one could accurately steal the model architecture and its parameters. Subsequent research (Xu et al., 2022) has thoroughly investigated the strategy of ensembling victim models to train a competitor model that surpasses its teachers.

The exploration of generative language model extraction is still in its infant stage, with only a handful of studies thus far. Wallace et al. (2020) investigate imitation attacks on natural language models. By designing monolingual query texts and collecting responses, they successfully extract the knowledge from a simulated machine translation model under the black-box settings. This research exhibits that slight architectural differences will not influence the extraction between language models. Li et al. (2023b) also explores the potential risks of stealing the code-generation abilities of LLMs into smaller downstream models. Unlike previous research (Wallace et al., 2020), this is the first study that selects LLMs as targets. By collecting large-scale domain-specific samples, they fine-tune a 7-billion local pre-trained model with them and show the similarity between the victim and local models in both performances and adversarial samples. However, these two studies employ the MLE loss (Equation 3) as the MEA method, neither considering whether MLE is compatible with LLMs's training, especially the alignment procedure shown in Section 2.2, nor addressing optimizations related to query efficiency and the watermark resistance. Besides, the scope of these studies is limited to stealing specific knowledge in a few downstream domains. At the same time, most of the critical

| Model/Metric | Accuracy | Precision | Recall | F1 Score |
|---|---|---|---|---|
| *PIQA (Bisk et al., 2020) with 64 query samples* | | | | |
| Victim Model | 0.828 | 0.828 | 0.827 | 0.827 |
| Local Model | 0.622 | 0.638 | 0.621 | 0.609 |
| +MLE (baseline) | $0.760 \pm 0.02$ | $0.771 \pm 0.01$ | $0.760 \pm 0.02$ | $0.757 \pm 0.03$ |
| +KD (gre-box) | $0.759 \pm 0.02$ | $0.760 \pm 0.02$ | $0.759 \pm 0.02$ | $0.759 \pm 0.02$ |
| +LoRD (ours) | $0.785 \pm 0.01$ | $0.795 \pm 0.01$ | $0.785 \pm 0.01$ | $0.783 \pm 0.02$ |
| *TruthfulQA (Lin et al., 2021) with 64 query samples* | | | | |
| Victim Model | 0.414 | 0.500 | 0.207 | 0.293 |
| Local Model | 0.391 | 0.500 | 0.195 | 0.281 |
| +MLE (baseline) | $0.381 \pm 0.17$ | $0.500 \pm 0.00$ | $0.190 \pm 0.09$ | $0.266 \pm 0.09$ |
| +KD (gre-box) | $0.463 \pm 0.03$ | $0.500 \pm 0.00$ | $0.232 \pm 0.01$ | $0.316 \pm 0.01$ |
| +LoRD (ours) | $0.408 \pm 0.05$ | $0.500 \pm 0.00$ | $0.204 \pm 0.03$ | $0.289 \pm 0.03$ |

Table 6: MEA comparison on QA tasks among MLE and our LoRD methods, where we use GPT-3.5-turbo as the victim model, and Llama3-8B (lla, 2024) as the local initial model.

aspects of LLMs and the required extraction capabilities, such as query numbers and local model scales, remain unresolved.

### C.3 TEXT WATERMARKS

In contrast to stealing LLMs, IP protection methods have received considerable attention recently. By sampling a stealthy but representative "greed word set" on the vocabulary distribution, these methods (Cong et al., 2022; He et al., 2022; 2021; Kirchenbauer et al., 2023) can remap the generated words into their synonyms or add the "watermarked" token automatically, and thus effectively certify the output. Besides, strategies such as integrating embeddings into the representation as the backdoor (Peng et al., 2023b) or manipulating the probabilities with crafted sinusoidal noises (Zhao et al., 2022; 2023) are also proposed. However, these approaches often presume more stringent conditions regarding the victim and the suspected models. This paper will further assess the effectiveness of LoRD and current MEAs in evading these black-box watermarking strategies.

## D A DETAILED THREAT MODEL

**Adversary's Objective.** The adversary's objective is to steal the targeted knowledge from LLMs. Specifically, we select machine translation, reasoning, data-to-text, structured text generation, and summarization as the downstream domain-specific tasks. The adversary aims to develop a *query-efficient* MEA algorithm, since the amount of input and generated tokens will be counted as the costs. Additionally, the MEA methods are expected to be *watermark-resistant*, i.e., they are highly desired to reduce the risks of exposure to unauthorized stealing.

**Targeted Models.** We select Llama3-70B, GPT-3.5-turbo, and GPT-4o as the victim models in this paper. Unlike previous works that only deployed simulated local victim models (e.g., OPT (Zhang et al., 2022)), our selections aim to expose the stealing threat on realistic AI services. Besides, our target models are specifically constrained to LLMs fine-tuned with alignment methods (e.g., RLHF) since they are not only state-of-the-art solutions now but also more valuable due to their human-based alignments.

**Adversary's Capabilities.** In accordance with the LLM-based AI service APIs, we identify two attack scenarios: black-box and grey-box attacks. In the black-box scenario, only textual responses the adversary is allowed to obtain. At the same time, all other information, such as the temperature, sampling strategies, and the hidden states of LLMs, are unseen and inaccessible. On the contrary, a grey-box attack allows the adversary to access the generation probabilities distribution of tokens. Notice that both MLE and our LoRD method are under black-box settings, and we only adopt grey-box settings on some particular stealing methods, such as knowledge distillation.

Besides, this paper posits that the adversary usually has worse training conditions than the victims. Specifically, query times and the scale of the local model available to the adversary are much smaller than the victims' training datasets and model parameters. This setting has been adopted in previous LLMs' extraction (Li et al., 2023b). We call it a LaViSH (**La**rge-**Vi**ctim-**S**mall-**H**eist) framework, which allows us to estimate the upper bound of MEA risks empirically. For adversaries with more

| Task | Instruction |
|---|---|
| WMT16 | Please translate the sentence from [source language] to English. |
| PiQA & TruthfulQA | Please select the correct answer for the "Question" of Users. Question: [question] Selection 1: [Selection1] Selection 2:[Selection2]. |
| E2E NLG | Please translate the information to a sentence in natural language. |
| CommonGen | Please generate a sentence based on the words provided by Users. |
| WikiSQL& Spider | Please return to me the SQL sentence based on the text (i.e., Question) and the table information (i.e., Table) provided by the User. |
| TLDR& SamSUM | Please **summarize** the content given by the user. |
| CNN Daily Mail | Please **summarize** the content given by the user. |

Table 7: Instructions used in the different downstream datasets.

substantial resources, they can train more powerful MEA-based LLMs by leveraging MEA algorithms under our LaViSH settings.

# E  LIMITATIONS AND FUTURE WORKS

**MEAs on Multi-modal Models.** While this paper delves into MEAs for large language models, it acknowledges the oversight of the multi-modal attribution of current commercial models (Anil et al., 2024; Achiam et al., 2024) that integrate various forms of data such as text, images, voice, and so on. The challenge of extending MEA algorithms to accommodate these models, which requires extra considerations on the unified representation of concepts, remains unexplored. Future work could focus on developing MEA methodologies sensitive to multi-modal data nuances.

**Capacities beyond LaViSH Settings.** We utilize the LaViSH setting to describe the model capacity of adversaries in our threat model (see Appendix D). However, sometimes, the adversary might possess comparable or superior training resources to the victims. Though this paper posits that our MEA algorithms and theoretical analysis are still compatible with such conditions, we concede that concrete experimental validation and results beyond LaViSH settings are not presented here.

**Lower-level Extractions.** This study evaluates MEAs at the performance level, i.e., it measures the extraction effectiveness simply through task performance metrics, or the similarity of learned distributions to the victim model. This setting is justified, as performance metrics are essential for evaluating task-related knowledge and the practical application of LLMs. However, it does not consider the lower-level similarities between the victim and local models. Can we achieve neuron-level alignments in LLM's MEAs? How does a LaViSH setting hurt LLM's MEAs? Is it compatible to extract a MoE (Mix-of-the-Expert) (Shazeer et al., 2017) victim model with a dense local model? These questions are not addressed in this research.

---

**Algorithm 1** LoRD Algorithm

---

1: **Input:** Query dataset $\mathcal{D}_q$, local language model $\theta_{init}$, interface of the victim model $P_{\theta_{vic}}(\cdot|\cdot)$, train period number $N_t$, threshold values $\tau_1$ and $\tau_2$.
2: // Initialization.
3: $\theta_0 \leftarrow \theta_{init}, \mathcal{D}_{tr} \leftarrow \emptyset, \mathcal{D}_0^+ \leftarrow \emptyset, \mathcal{D}_0^- \leftarrow \emptyset, t \leftarrow 0$;
4: // Query the victim model.
5: **for** $\mathbf{x} \sim \mathcal{D}_q$ **do**
6: $\quad \mathbf{y}_{vic} \leftarrow P_{\theta_{vic}}(\cdot|\mathbf{x})$;
7: $\quad \mathcal{D}_{tr} \leftarrow \mathcal{D}_{tr} \cup \{(\mathbf{x}, \mathbf{y}_{vic}, P_{\theta_{vic}}(\mathbf{y}_{vic}|\mathbf{x}))\}$;
8: **end for**
9: // Train local model.
10: // Initialize the positive and negative datasets.
11: $\mathcal{D}_0^+ \leftarrow \mathcal{D}_q$;
12: **for** $(\mathbf{x}, \mathbf{y}_{vic}, P_{\theta_{vic}}(\mathbf{y}_{vic}|\mathbf{x})) \sim \mathcal{D}_{tr}$ **do**
13: $\quad \mathbf{y}_0^- \sim P_{\theta_t}(\cdot|\mathbf{x})$;
14: $\quad \mathcal{D}_0^- \leftarrow \mathcal{D}_0^- \cup \{(\mathbf{x}, \mathbf{y}_0^-, P_{\theta_0}(\mathbf{y}_0^-|\mathbf{x}))\}$;
15: **end for**
16: **while** $t < N_t$ **do**
17: $\quad t \leftarrow t + 1$;
18: $\quad \theta_t \leftarrow \theta_{t-1}$;
19: $\quad$ **for** $(\mathbf{x}, \mathbf{y}_{vic}, P_{\theta_{vic}}(\mathbf{y}_{vic}|\mathbf{x})) \sim \mathcal{D}_{tr}$ **do**
20: $\quad\quad \mathbf{y}_t^+, \mathbf{y}_t^- \sim P_{\theta_t}(\cdot|\mathbf{x})$;
21: $\quad\quad \mathcal{D}_t^+ \leftarrow \mathcal{D}_t^+ \cup \{(\mathbf{x}, \mathbf{y}_t^+)\}$;
22: $\quad\quad \mathcal{D}_t^- \leftarrow \mathcal{D}_t^- \cup \{(\mathbf{x}, \mathbf{y}_t^-)\}$;
23: $\quad$ **end for**
24: $\quad$ // Forward.
25: $\quad$ **for** $\mathbf{x}, \mathbf{y}_{vic}, \mathbf{y}_{t-1}^+, \mathbf{y}_{t-1}^- \sim (\mathcal{D}_{tr}, \mathcal{D}_{t-1}^+, \mathcal{D}_{t-1}^-)$ **do**
26: $\quad\quad \Delta^+ \leftarrow \log P_{\theta_t}(\mathbf{y}_{t-1}^+|x) - \log P_{\theta_{t-1}}(\mathbf{y}_{t-1}^+|x)$;
27: $\quad\quad \Delta^- \leftarrow \log P_{\theta_t}(\mathbf{y}_{t-1}^-|x) - \log P_{\theta_{t-1}}(\mathbf{y}_{t-1}^-|x)$;
28: $\quad\quad$ **if** $\Delta^+ < \Delta^-$ **then**
29: $\quad\quad\quad$ swap$(\mathbf{y}_{t-1}^+, \mathbf{y}_{t-1}^-)$;
30: $\quad\quad\quad$ swap$(\Delta^+, \Delta^-)$;
31: $\quad\quad$ **end if**
32: $\quad\quad$ **if** $P_{\theta_t}(\mathbf{y}_{t-1}^+|x) < \tau_1$ && $\Delta^+ < \tau_2$ **then**
33: $\quad\quad\quad \mathbf{y}_{t-1}^+ \leftarrow \mathbf{y}_{vic}$;
34: $\quad\quad$ **end if**
35: $\quad\quad$ // Compute loss with Equation 10 or 11.
36: $\quad\quad \mathcal{L} \leftarrow \log[\frac{P_{\theta_t}(\mathbf{y}_{t-1}^-|\mathbf{x})}{P_{\theta_t}(\mathbf{y}_{t-1}^+|\mathbf{x})}] + clip(\log[\frac{P_{\theta_t}(\mathbf{y}_{t-1}^-|\mathbf{x})}{P_{\theta_t}(\mathbf{y}_{vic}|\mathbf{x})}])$;
37: $\quad\quad$ // Backward.
38: $\quad\quad \theta_t \leftarrow \text{stepUpdate}(\theta_t, \mathcal{L})$;
39: $\quad$ **end for**
40: **end while**
41: **return** $\theta_t$

---

