# OpenReview forum: "Alignment-Aware Model Extraction Attacks on Large Language Models"
_ICLR.cc/2025/Conference — Submitted to ICLR 2025_

### Official Review · Reviewer_5Msi · 2024-10-27

**Soundness:** 2
**Presentation:** 4
**Contribution:** 2
**Rating:** 5
**Confidence:** 4

**Summary:**

This paper proposes a new model extraction attack (MEA) algorithm, named LoRD. The authors claim that existing MEA methods suffer from not taking the preference alignment process into consideration during stealing. The authors try to use the victim model's response as the guidance to help select the local chosen and rejected responses (also as the optimization target in some cold start cases and regularization terms). The authors believe their loss design can make the attack more efficient and resistant to text watermark defenses.

**Strengths:**

1. The studied problem is practical and meaningful. The motivation is good and reasonable.

2. The paper is well-written, the tables and figures are well displayed. I thank the authors for their great efforts on well organizing their manuscript.

3. I like the analysis about Figure 5, which shows some novel and interesting insights.

**Weaknesses:**

I have some concerns about current submissions. I hope the authors could address my questions below. I may adjust my scores based on the authors' responses.

1. First, I should suggest the authors to carefully read and follow the Author Guide in the website to place the Ethical Statement (and optionally the Reproducibility Statement) after main text before references, rather than in the Appendix.

2. Regarding the method part, in Line 210-211, the authors states that "indicate whether a selected sentence is locally isotropic to the victim model’s response... in the current optimization step". However , from Figure 3 and Eq. (8) we can see that the victim model's response $y_{vic}$ is not used in deciding chosen ($y^{+}$) and rejected ($y^{-}$) responses and in objective loss function $L_{obj}$ (unless in the cold start case). So I am wondering how the victim model's response can guide the preference alignment of the local/target model?

3. Regarding the form of objective function in Eq. (8), it seems to be very similar to SimPO [1] loss (without some regularizations). So I am wondering why do authors not try this straightforward idea to perform model stealing: sample two responses from local model, prompt the victim model to directly decide the chosen one and rejected one, then places them into the Eq. (8).

4. Regarding the regularization term, it has the same form of the objective function but places $y^{+}$ with $y_{vic}$ in the denominator. I think the function of this regularization function if to consider the victim model's response the chosen response and directly distill the knowledge of victim model into the local model. So why call it the regularization function?

5. I do not fully understand the analysis about Query Efficiency in Section 4.2. I am confused why the ideal query times for LoRD can be reduced to $O(V^{NQ}\times C)$.


6. The Watermark Resistance part is interesting and reasonable. But I think selecting vocabulary-splitting based watermarking method [2] is inappropriate (as we can see, the p-values of MLE are already very high), the authors should choose backdoor-based watermarking methods [3,4], which would make the results more convincing.

7. The experimental results in Table 1 show limited improvement over baseline MLE.


[1] Meng, Yu, Mengzhou Xia, and Danqi Chen. "Simpo: Simple preference optimization with a reference-free reward."

[2] Kirchenbauer, John, et al. "A watermark for large language models." ICML 2023

[3] Gu, Chenxi, et al. "Watermarking pre-trained language models with backdooring."

[4] Li, Peixuan, et al. "Plmmark: a secure and robust black-box watermarking framework for pre-trained language models." AAAI 2023

**Questions:**

There are some typos or presentation errors:

(1) In Eq. (4), $y_{i, <j}$ should be bolden.

(2) In Eq. (6), why $\hat{y}$ in the first term but $y$ in the second term.

(3) Figure 2, Step 4, should "and" be "or" (according to the last line in Page 4)?

(4) In Appendix, there are a lot of misuses of ```\citet``` (should be ```\citep```).

**Details Of Ethics Concerns:**

Please place your Ethical Statement after the main text before references as required by ICLR guidelines.

---

> ### Author Response · Authors · 2024-11-23
>
> Thanks for your in-depth reviews. We have revised our paper based on your feedback, to address the following points:
>
> 1. We have corrected the placement of the ethical statement.
> 2. We have fixed typos and errors related to your questions.
> 3. We have reorganized Tables 1, 2, and 5 to provide a more comprehensive and fair understanding of the performance.
> 4. We have rewritten the methodology section for clarity.
> 5. We have appended an explanation of the query time complexity for LoRD.
>
> Here are our point-by-point responses to your questions:
>
> - Line 210-211: The victim model's response is introduced in the regularization term, which influences the optimization direction of the local model and, consequently, the selection of positive and negative models.
> - Why not SimPO: We are aware of these methods, but they require an extra prerequisite of the victim model, i.e., the ability to judge responses beyond simply accomplishing the given task. This prerequisite cannot always be fulfilled, due to limitations in the victim model's capabilities and the stealthy requirement of model extraction. Additionally, they are actually more complex for the adversary, as they involve issues such as prompt design and victim model capabilities. Despite these limitations, we admit that one could potentially improve the effectiveness of LoRD by more accurately judging positive samples.
> - Regularization term: Thanks again for your thoughtful insights. We agree that "regularization" is not the most appropriate term here. We used it following the tradition in RL, where the objective function is about the exploration (i.e., learning) of the local model, and the regularization term limits the local model from deviating too far from the victim model's response. This is the physical meaning of "regularization". While in RL the regularization constrains the optimized model with the last-step-optimized model, in MEA we constrain the $y^+$ with $y_{vic}$. The both are designed to ensure convergence.
> - Complexity of query times: an intuitive explanation is shown in Figure 11. Formally, we need $O(V^{N\_Q})$-level query samples to represent the input data distribution. Since there is usually not just one correct answer in generative tasks, we can consider that there are at most $O(V^{N\_R})$ responses per query, which is the complexity for a token-level MLE. LoRD does not affect the query side, so the required query times to represent the whole input data distribution remains unchanged. However, ideally, LoRD can find all responses that are "isotropic" to the victim model from the search in $(V^{N\_R})$ candidates. That's the reason we refer to it as $O(1)$ complexity on the generation side. In realistic scenarios, it is impossible to find all candidates from a single "standard answer", so we relax $O(1)$ to $O(C)$ where the value $C$ quantifies the capability of the initial local model and thus should be considered as a constant.
> - Watermark Resistance Experiments. Thanks for your suggestions on defenses. Regarding model-level watermarks, we have already discussed them in our potential defenses section. We believe that model-level watermarks are effective against both LoRD and MLE once the local model learns about triggers (backdoors). We have further detailed this part based on the literature you provided in our revised version. However, we face challenges in fulfilling your requirements: i) unlike content-level watermarks which has both packages and official implementations [1], model-level watermarks are still in their academia stage, making them challenging for experiments; ii) current academic studies in this field (including the two you list) have only tested their efficacy on BERT, while the effectiveness on generative and generalized NLP tasks is not clear; iii) as we mentioned in the paper, model-level watermarks will become ineffective if the query set of the adversary does not cover the backdoor triggers. This situation is common, as it is difficult for a generalized LLM to possess backdoors in each downstream field. In summary, in the raw paper we have acknowledged that model-level watermarks are effective and are a promising direction to mitigate our proposed methods, while we also discussed their limitations as well as their research states.

---

> > ### Comment · Reviewer_5Msi · 2024-11-23
> > **Response**
> >
> > Thank you for your clarifications. Some of my questions have been addressed. However, there are still some concerns remaining. First, I can now understand the meanings of Line 210-211 and the regularization term. But I am not convinced by your clarification that "issues such as prompt design and victim model capabilities". The prompt can be a simple llm-as-a-judge prompt, and the capabilities of current instruct-models should be enough to do this task. I would expect to see an empirical comparison with this baseline. It is not direct for me to see that the current method which implicitly uses $y_{vic}$ to identify chosen and rejected pairs is better. Second, the concern on the limit empirical improvement still exists, which is also pointed by other reviewers.
> >
> > Based on the current response, I may raise my score to a 5, after discussing with other reviewers.

---

> > > ### Author Response · Authors · 2024-11-23
> > >
> > > Thanks for your timely feedback.
> > >
> > > We appreciate your potential promise of improving the score and acknowledge the remaining concerns you have raised. Here, please permit me to explain these two concerns for you again:
> > >
> > > 1. **Why not a direct feedback from the victim model?** Based on your response, we fully understand and accept your rebuttal on the complexity of a prompt-based direct feedback in victim models. However, we maintain our stance that such a approach may not be suitable in some realistic stealing scenarios for the following reasons:
> > >  - *i)* A direct feedback query will *expose the intention of the adversary*;
> > >  - *ii)* Unlike the current design of LoRD, direct feedback is contingent upon the local model's responses, which is *query-inefficient*. Specifically, for a given query sample, the algorithm would need to repeatedly query the victim model to distinguish between $\mathbf{y}^+$ and $\mathbf{y}^-$ across different learning periods. On the contrary, LoRD necessitates only a single query per sample to discriminate different $(\mathbf{y}^+,\mathbf{y}^-)$ pairs;
> > >  - *iii)* The *threat model changes* when empolying this strategy. Both LoRD and MLE are currently trained under the same conditions, i.e. $(\mathbf{x},\mathbf{y}_{vic})$ paires. The fairness would be questioned when we compare methods under disparate query settings.
> > >  - Nevertheless, we'd like to append some empirical experiments of introducing the direct feedback into LoRD and also, add some baselines. We hope we have the time to accomplish these two experiments.
> > > 2. **Limited empirical improvements.** We would like to clarify that the improvements compared to MLE are not limited. We evaluate LoRD under 5 downstream tasks, achieving 2\~3 points of improvements on QA (Table 5), 1\~4 points of improvements on machine translation (Table 2), and 2 points of improvements on 2/3 datasets of Summarization (Table 1). For the other two tasks, actually LoRD still outperforms MLE when given smaller query numbers. The problem lies in our experiment results organization: *we placed the three tasks where LoRD performed worst in the main table at the beginning.* Some other experiments in the paper, such as the query efficiency and the model scale's influence shown in Figure 7 and Figure 8, can also support the effectiveness of LoRD. Despite it, it is not necessary for LoRD to outperform MLE in all scenarios and all metrics. For those tasks which can be easily generalized or learned, both LoRD and MLE may reach the cellar together, which exhibits a comparable experimental result.
> > >
> > > In summary, we provide explanations for the remaining concerns, and will make further revisions to the paper in a few days, including:
> > > 1. re-organizing the experiment results among Table 1, 2, and 5.
> > > 2. discussing, revising, or comparing LoRD with SimPO.
> > >
> > > We appreciate your suggestions again, and look forward to your continued feedback.

---

> ### Author Response · Authors · 2024-11-28
> **Comparison with SimPO**
>
> Based on your response, we compare LoRD with SimPO. SimPO incorporates two labels, $y\_w$ and $y\_l$, representing the winner and loser of the generated texts, respectively. Specifically, we reproduce SimPO according to Equation (6) outlined in SimPO's paper, and adopt the recommended hyper-parameter settings from SimPO's source code, which entails setting $\beta$ to 2.5 and $\gamma$ to 1.375.
>
> It is important to note that SimPO was not originally designed for model extraction tasks but rather as a candidate for DPO, and we have made necessary modifications. We adpat SimPO for model extraction task, considering the following two implementations:
> 1. SimPO-I: Assigns $y^+$ as $y\_w$, and $y^-$ as $y\_l$.
> 2. SimPO-II: Utilizes $y\_vic$ as $y\_w$ and $y^-$ as $y\_l$.
>
> Here is the performance comparison on WMT (de-en):
>
> | Method      | BLEU-1 | BLEU-2 | BLEU-3 | BLEU-4 | BERTScore-Pre | BERTScore-Rec | BERTScore-F1 | Rouge-L-F1 |
> |-------------|--------|--------|--------|--------|---------------|---------------|--------------|------------|
> | LoRD (ours) | 54.40  | 42.18  | 33.56  | 27.06  | 89.09         | 94.06         | 91.44        | 56.09      |
> | SimPO-I     | 29.25  | 19.95  | 14.93  | 11.59  | 83.32         | 88.67         | 85.85        | 30.04      |
> | SimPO-II    | 35.47  | 25.12  | 19.12  | 14.86  | 86.42         | 90.21         | 88.22        | 34.77      |
>
> Based on the experimental results, it is evident that both SimPO methods exhibit inferior performance compared to LoRD within the same query budget. This observation indicates that SimPO may not be a highly query-efficient approach for model extraction attacks.

---

> ### Comment · Reviewer_5Msi · 2024-12-03
>
> I appreciate the authors' additional results. However, what I expected is to prompt the victim model to decide the chosen and rejected responses, instead of directly using the chosen and rejected responses decided by LoRD to perform SimPO. I do think the responses addressed part of my concerns, and I would increase my score to a 5. I invite the authors to incorporate all the additional results and suggestions from other reviewers into the future version or submission. Thank you.

---

> ### Author Response · Authors · 2024-12-03
> **Empirical Exploration on the Direct Prompt based Stealing**
>
> We sincerely thank you for your valuable response and constructive suggestions.
> Most of the experiments and revisions have been incorporated into the revised version of our paper, and we remain dedicated to addressing all concerns raised by the reviewers and will not withdraw our submission before the final decision.
> We kindly hope that the reviewer might reconsider the score if the explanation address the remaining concern.
>
> In response to your concern about why we believe *direct feedback from the victim model is not ideal for MEAs*, we acknowledge that our previous explanation may not have been sufficient in the experiment part. To address this further, we have conducted additional experiments, as detailed below.
>
> **Settings.**
>
> We designed the prompt for obtaining feedback as follows:
>
> ```
> For a translation task involving the conversion of the given `Text` into English, the user will provide two translation versions labeled `A` and `B`. Your task is to return the *letter corresponding to the better translation* without including any additional output.
> ```
>
> In each training step, the local model generates two candidate responses. Using the above instruction, we determine the positive response, which is then used along with the negative response to fine-tune the local model under the SimPO loss function. We maintain the usage of hyperparameters in the previous responses.
>
> **Experiment results.**
>
> The experimental results are as follows:
>
> | Selected Local Model | BLEU-1 | BLEU-4 | BLEU-4 | BLEU-4 | BERTScore (Pre) | BERTScore (Rec) | BERTScore (F1) | Rouge-L (F1) |
> |----------------------|--------|--------|--------|--------|-----------------|-----------------|----------------|--------------|
> | LoRD (ours) Q=16     | 54.40  | 42.18  | 33.56  | 27.06  | 89.09           | 94.06           | 91.44          | 56.09        |
> |----------------------|--------|--------|--------|--------|-----------------|-----------------|----------------|--------------|
> | SimPO (T=1.0) Q=16   | 44.80  | 34.80  | 27.94  | 22.83  | 89.79           | 93.50           | 91.57          | 48.39        |
> | SimPO (T=1.3) Q=16   | 44.19  | 33.45  | 26.31  | 21.18  | 88.49           | 92.65           | 90.47          | 47.09        |
> | SimPO (T=0.8) Q=16   | 42.99  | 31.81  | 24.85  | 19.82  | 90.37           | 88.32           | 92.64          | 44.04        |
> |----------------------|--------|--------|--------|--------|-----------------|-----------------|----------------|--------------|
> | SimPO (T=1.3) Q=256  | 3.09   | 0.13   | 0.00   | 0.00   | 68.04           | 81.54           | 74.17          | 11.22        |
> | SimPO (T=0.8) Q=256  | 20.99  | 10.75  | 7.01   | 5.04   | 85.56           | 87.52           | 86.50          | 21.08        |
>
> In the table, `T` denotes the sampling temperature, and `Q` denotes the query times.
>
> **Analysis.**
>
> We conducted experiments with various sampling temperatures, yet the efficacy of stealing remained constrained under identical settings. This limitation may stem from the local model's lack of guidance from *correct answers*. When the local model generates two suboptimal responses, a direct prompting-based method is compelled to select the "winner" of two inadequate response rather than an optimal response, which we believe is the crux of the issue.
>
> RLHF tackles this challenge by incorporating a regularization term with the initial model, LoRD addresses it through our $L_{reg}$, leveraging the victim model's response, and DPO resolves it by employing the training corpus of the reward model. unfortunately, a direct prompt-based method overlooks this point.
> To further investigate this problem, we increased the query number to 256, which resulted in the local model failing to converge and exhibiting poor performance.
>
> Besides, we also observed **a bias in the victim model's selection** between the first and second sentences. In a series of 256 queries, the model successfully provided an answer (either A or B) 255 times. However, it chose the first sentence only 84 times, which is a mere 32.94%, significantly deviating from the expected 50%. Given that the generated sentences are randomly sampled from the local model without any significant correlation to their order, we deduce that relying on the victim model to directly generate feedback might be, at best, an unreliable approach. It may necessitate additional considerations for the design of the prompt and the capabilities of the victim model to ensure robustness.
>
>
> We hope that the above empirical explanation further addresses your concerns. If not, we would be delighted to engage in further discussion after the review process, if possible.
>
> We sincerely appreciate all of your previous feedback and suggestions. Thank you!

---

### Official Review · Reviewer_aNx7 · 2024-11-01

**Soundness:** 3
**Presentation:** 3
**Contribution:** 3
**Rating:** 6
**Confidence:** 4

**Summary:**

This paper designs a new model extraction attack targeting LLMs. The method innovatively uses reinforced distillation, allowing a local model to more quickly and accurately learn the victim model’s knowledge. Moreover, thanks to reinforcement learning, the local model does not learn the watermark embedded in the victim model. The authors conducted extensive experiments to verify the effectiveness of this method.

**Strengths:**

- The paper is well-written with a clear structure and rich content, making it easy to follow.
- The authors designed a new reinforcement learning-based distillation method called Locality Reinforced Distillation (LoRD), achieving better results in LLM model extraction problems.
- The method can steal commercial models under a fully black-box threat model, making it highly practical.
- Unlike supervised learning-based methods (MLE), LoRD does not imitate the victim’s generation as labels, so it does not replicate possible watermarks in the victim model.
- LoRD’s learning approach improves the way LLMs are extracted, thereby reducing the cost of queries.
- Although the method is not highly effective on every task, the authors have deeply explained the reasons behind these issues.
- As an attack method against LLMs, the authors responsibly discussed ethical concerns and provided some possible defense strategies.

**Weaknesses:**

- The method is a domain-specific model extraction method; the authors should clarify this in the introduction section.
- The design of the method includes some thresholds. Although the authors provided specific values, they did not carefully introduce the impact of these parameters and whether attackers need to set different thresholds for different local and victim models.
- In Equation 8, the authors removed a term but did not explain the deeper reasons and impacts.

**Questions:**

- In Equation 8, does removing P_{y_{vic}} have negative effects?
- In Equation 9, if using y+ instead of y-, what differences and impacts would there be?
- For some commercial models that do not provide generation probabilities, how effective is this method?

---

> ### Author Response · Authors · 2024-11-23
>
> Thank you for your valuable feedback. We are currently conducting four experiments to address your questions, and we hope to provide the results before the rebuttal period ends.
>
> Regarding your final question, we have only evaluated current commercial LLMs on domain-specific tasks, as shown in Figure 9.
>
> Thank you.

---

> ### Author Response · Authors · 2024-11-26
> **Experiments**
>
> ### Experiments beyond Domain-specific Stealing
> For weakness 1, we list conducted the experiments on safety alignment extraction. Specifically, we utilized two open-source datasets for these experiments, namely SafeRLHF and DiaSafety, to assess the safety of the responses generated. We employed PerspectiveAPI to automatically evaluate the safety of the responses. The API identifies five key aspects of safety probabilities: Toxicity, Insult, Profanity, Severe Toxicity, and Threat. In these categories, a lower score indicates better safety performance.
> For the LoRD model, we have retained the same hyper-parameters as those used in our domain-specific experiments to ensure consistency.
>
> **DiaSafety**:
>
> | Model               | Toxicity(%) | Insult(%) | Profanity(%) | Severe Toxicity(%) | Threat(%) |
> |---------------------|-------------|-----------|--------------|--------------------|-----------|
> | Llama3-8B (initial) | 14.20       | 7.94      | 8.35         | 1.58               | 2.29      |
> | Llama3-8B + MLE     | 8.31        | 3.69      | 4.31         | 0.83               | 1.50      |
> | Llama3-8B + LoRD    | **6.45**    | **2.81**  | **3.56**     | **0.71**           | **1.34**  |
>
>
>
> **SafeRLHF**:
>
> | Model            | Toxicity(%) | Insult(%) | Profanity(%) | Severe Toxicity(%) | Threat(%) |
> |------------------|-------------|-----------|--------------|--------------------|-----------|
> | Llama3-8B        | 7.92        | 2.71      | 2.80         | 0.30               | 1.49      |
> | Llama3-8B + MLE  | 4.87        | 1.98      | **1.66**     | **0.16**           | 1.02      |
> | Llama3-8B + LoRD | **3.55**    | **1.15**  | 2.84         | 0.38               | **0.79**  |
>
>
> ### Ablation Study
>
> We also conducted an ablation study to compare several variants of LoRD for your questions 1 and 2. We applied the same LoRD settings to the WMT (de-en) dataset and compared LoRD's performance with the variants you mentioned.
>
>
> | Method            | BLEU-1 | BLEU-2 | BLEU-3 | BLEU-4 | BERTScore-Pre | BERTScore-Rec | BERTScore-F1 | Rouge-L-F1 |
> |-------------------|--------|--------|--------|--------|---------------|---------------|--------------|------------|
> | LoRD              | 54.40  | 42.18  | 33.56  | 27.06  | 89.09         | 94.06         | 91.44        | 56.09      |
> | w. $y_{vic}$ (Q1) | 55.26  | 42.57  | 33.61  | 27.01  | 89.27         | 94.14         | 91.57        | 56.18      |
> | use $y^+$ (Q2)    | 52.16  | 40.33  | 32.06  | 25.87  | 87.41         | 93.28         | 90.19        | 54.12      |
>
> Conclusion of the Ablation Study:
>
> 1. For Question 1: Removing the term $y_{vic}$ slightly reduces the performance in the stealing task, with the decrease being less than 1 point. Threfore, **as we have done in the paper, we can conclude that this term can be omitted without significantly impacting the results.**
> 2. For Question 2: **Our approach, which incorporates $y^-$ in the regularization term, yields better performance** compared to replacing $y^-$ with $y^+$. This indicates that the use of $y^-$ is more effective for our design objectives.

---

> > ### Comment · Reviewer_aNx7 · 2024-12-03
> > **Response to the rebuttal**
> >
> > Thanks for the experiments' efforts and the authors' explanations. The rebuttal addressed most of my concerns, and I will keep my positive attitude on this work. Good luck!

---

> > > ### Author Response · Authors · 2024-12-03
> > >
> > > Thank you for your positive attitude of our work. We really appreciate it. Thank you!

---

### Official Review · Reviewer_Bcxr · 2024-11-02

**Soundness:** 1
**Presentation:** 2
**Contribution:** 2
**Rating:** 3
**Confidence:** 4

**Summary:**

This paper proposes a new model stealing attack, particularly geared towards target LLMs that are aligned via RLHF.

**Strengths:**

1. The premise of the paper seems promising and does try to fill in an important gap in the knowledge about model stealing attack against modern LLM, particularly targeting the RLHF process. This seems like an interesting problem with potential impact.

**Weaknesses:**

1. **Lack of rigor and explanation in the derivation of the loss functions**
    1. L228: I'm quite lost here about why this is the right derivation of $R_{\theta_\phi}$ from Eq. 6. I'm not an expert in RL so I may just miss something obvious, but I'd like to see the full derivation somewhere in the paper.
    2. L239: These design decisions seem rather unprincipled to me. The scaling term is simply dropped. To make a claim that the algorithm still works as intended after the approximation, I'd like to see an ablation study to validate this point.
    3. L248 (”requires an extra exponential…”): I'm a bit confused by this claim. I'm not sure why just one more exponentiation would noticeably increase the runtime to the point that it is a consideration for the attacker.
    4. L250 (”only the logarithmic part of KL divergence is employed”): Again, this seems quite unprincipled. What effect does it have? Any ablation study?
    5. L251: What are the "selected tokens" in this case?
    6. Eq 9: Here, the KLD term is not even applied to the current and the initialized models. It is only on the current model with two different outputs (they are not distributions). It is not KLD anymore. I do not understand why it is motivated by the usual KLD term or whether they even serve the same purpose.
    7. L264 (”Finally, we wrap L_LoRD with a sigmoid…”): Why is this necessary?
2. **Lack of rigor in theoretical analysis in Section 4**
    1. The theoretical analysis unfortunately lacks any rigor or real purpose in the paper. Proposition 1 is not a well-defined mathematical statement. While the proofs in the appendix are "not wrong," they add no information and do not support this proposition. Proposition 2 also does not support the proposed method.
    2. Section 4.2 also lack mathematical rigor, and the statements are handwavy. For example, the analysis on the number of queries for both MLE and LoRD seems to lack any derivation or source.
3. **Experiments**
    1. Why not evaluate the alignment of the model since the attack tries to imitate the RLHF process which is mostly used for safety training? If we are simply evaluating specific downstream tasks or knowledge, then using MLE to steal the model seems perfectly fine, and there is no need to use any alignment technique.
    2. The empirical results overall are relatively weak; LoRD almost has no improvement over the MLE baseline.
4. **Presentation**
    1. Figure 3: I'm not entirely sure what this figure is trying to communicate or add more information beyond the text or Figure 2. I might just be missing the point here.
    2. L205 (second paragraph of Section 3.1): This entire paragraph just dives into the technical design of the algorithm. I think it might be a good idea to just explain the intuition or the design choices in words before providing all the details.
    3. Table 1: I believe there are too many unnecessary numbers in this table. For example, perhaps only report F1 instead of precision and recall?

### Nitpicks

1. L30 (”ChatGPT cha (2024)”): There seem to be multiple typos on the citations throughout the paper.
2. There is a mistake where the authors cite references in the wrong format without parentheses, i.e., using `\citet` instead of `\citep` when using `natbib`. This happens so often that it slightly disrupts the reading.

**Questions:**

--

---

> ### Author Response · Authors · 2024-11-23
> **Response [1/3] Methodology Part**
>
> Thanks for your detailed reading and thoughtful review. We have carefully revised the paper in response to your concerns and provide point-by-point explanations below.
>
> ### The Design of Loss Functions
>
> Our target is to design an RL-style loss function for MEA, as shown in Equation (7). It consists of two parts, $L\_{obj}$ which represents the objective function, and $L\_{reg}$ which represents the regularization term. Equation (7) aligns with LLM's alignments (Equation (6)), where $L\_{obj}$ and $L\_{reg}$ correspond to $R\_{\theta\_{phi}}$ and $D_{KL}$, respectively. It is **not necessary** to design a loss function that is totally the same as or derived from LLM's RLHF, because there are various RLHF methods and their variants. Besides, in RL and RLHF, many methods, such as PPO, TRPO, and SimPO, often lack rigorous formal deductions beyond intuitive design. Nevertheless, we still aim to ensure that *{i)}* LoRD's loss converges consistently with LLMs' alignment, and *ii)* it converge at all. Our responses to your detailed comments are:
>
> - **Explanation of L228**: In RLHF, a reward model is typically trained to estimate the debiased reward of a sample $(x, \hat{y})$. This reward model is trained using the loss function defined in Equation (5). In LoRD, we do not train such a reward model. Instead, we "use the logarithmic proportion between the positive and negative samples as the debiased reward," following the definition in Equation (5). We provide a less formal deduction in the Appendix; however, it is important to emphasize that many related works (e.g., SimPO) did not provide rigorous justifications.
>
> - **"The scaling term is simply dropped":** Based on your feedback, we conducted an ablation study and revised the paper accordingly. Intuitively, dropping the scaling term does not significantly impact the efficacy of stealing because this term, introduced in PPO, primarily scales the reward to control the speed of convergence.
>
> - **About KL divergence:** We omit this term because our experiments demonstrated that it is neither efficient nor stable for model extraction tasks. An ablation study for this term is included in the paper.
>
> - **Explanation of L248:** We intended to show that $\log P$ is more natural than $P$ for implementation purposes. The former expands to $\log(\text{softmax}(\text{logits}))$, with $\log\text{softmax}$ being a more fundamental operator in modern ML frameworks.
>
> - **Clarification of line 251:** The term "selected tokens" refers to those tokens sampled during the generation process.
>
> - **Sigmoid function:** We provide an ablation study for this term. Our results indicate that $\text{sigmoid}$ serves a similar role to the `clip` term mentioned in the paper. While it is not strictly necessary, we recommend including it to enhance the stability of training.
>
> | Method              | BLEU-1 | BLEU-2 | BLEU-3 | BLEU-4 | BERTScore-Pre | BERTScore-Rec | BERTScore-F1 | Rouge-L-F1 |
> |---------------------|--------|--------|--------|--------|---------------|---------------|--------------|------------|
> | LoRD                | 54.40  | 42.18  | 33.56  | 27.06  | 89.09         | 94.06         | 91.44        | 56.09      |
> | w. $y_{vic}$ (W1.2) | 55.26  | 42.57  | 33.61  | 27.01  | 89.27         | 94.14         | 91.57        | 56.18      |
> | w. KL (W1.4)        | NC     | NC     | NC     | NC     | NC            | NC            | NC           | NC         |
> | w.o. Sigmoid (W1.7) | 50.01  | 37.73  | 29.65  | 23.77  | 89.25         | 93.73         | 91.38        | 50.39      |
>
> NC denotes not converged in our experiments.
>
> We appreciate your suggestions, which have made this paper clearer and provided it with a stronger motivation.

---

> ### Author Response · Authors · 2024-11-23
> **Response [2/3] Theoretical Analysis & Experiments**
>
> ### Theoretical Analysis
>
> 1. We acknowledge that Proposition 1 doesn't have a strict mathematical statement, which is why we call it a "proposition" rather than a "theorem". Despite this, we still provided an intuitive explanation (Figure 4) and in-depth analysis of four loss functions' optimization process (Appendix D.1) to support it. *While it is challenging to mathematically and precisely model and compare the converging procedures of neural networks under different loss functions, we made our best efforts to provide theoretical-level analysis and explanations beyond intuition and empirical experiments.*
> 2. "Proposition 2 does not support the proposed method". The objective of this research is not merely a new MEA method, and instead we aim to address a fundamental question that has not been investigated, i.e., whether MLE can be used to steal an RL-aligned LLM, together with its upper bound and limitations. Proposition 2 provides the answer that "yes, MLE can be used to steal LLMs and will reach the performance of the victim model when its loss function reduces to zero", where this proposition itself is one of the contributions of our paper. In addition, it is Proposition 2 that leads us to dig deep about the intrinsic strength of LoRD from Proposition 1, which lead to our analysis in Section 4.2.
> 3. "Lack of rigor in Section 4.2". To enhance the rigor of our analysis, we have appended an extra explanation part for Section 4.2, and cited some recent studies to support our analysis.
>
> ### Experiments
>
> - About Limited Improvements. **(i) The improvements compared to MLE are not limited.** We evaluated LoRD on five downstream tasks, achieving improvements of 2–3 points on QA (Table 5), 1–4 points on machine translation (Table 2), and 2 points on two out of three summarization datasets (Table 1). For the other two tasks, LoRD still outperforms MLE under smaller query numbers. *The issue lies in how our results were presented: we placed the three tasks where LoRD performed worst in the main table at the beginning, and in only half of these datasets did LoRD outperform MLE.*
> **(ii) LoRD doesn't aim to outperform MLE in all scenarios.** As shown in Proposition 2, LoRD and MLE can converge to the same endpoint when provided with sufficient query samples. Therefore, for tasks that are easily generalized or learned, both methods may reach the performance ceiling, resulting in comparable outcomes. The effects of query efficiency and model scale are detailed in Figures 7 and 8.
> - About Task Selection. We have included safety alignment experiments, as detailed in our responses below. While we agree that *safety alignment is essential and should be included in our experiments*, we also emphasize that *alignment is not merely about safety—it also significantly impacts task completion performance.* Therefore, the domain-specific evaluations presented in this paper remain valuable for comparing the performance of the two MEA methods.

---

> ### Author Response · Authors · 2024-11-23
> **Resposne [3/3] Presentation**
>
> ### Presentation
>
>
> 1. Figure 3 aims to intuitively exhibit how we select potentially positive and negative samples and why such a selection strategy is reasonable. It illustrates step 3 in Figure 2. The core idea is that we consider a generated sample as a positive sample if it has a higher increment in terms of model's familiarity after model optimization, which is also the meaning of "Locality Reinforcement".
> 2. Thanks for your suggestion on the intuitive explanation before details. We have added an explanation on intuition to the methodology following your advice.
> 3. Table 1: Following your feedback, we removed Rouge-L's Precision and Recall. We save BERTScore's all three metrics because the results can reflect why local models underperform the victim model.
>
> ### Typos
>
> 1. ”ChatGPT cha (2024)”: this seems the standard formation when citing urls in the `natbib` format.
> 2. We have checked and modified all incorrect citation formats in the revised version. Thank you.

---

> ### Author Response · Authors · 2024-11-26
> **Safety Alignment Exraction**
>
> ### Experiments of Safety Alignments
>
> In response to your feedback, we have conducted safety alignment experiments.
>
> We utilized two open-source datasets for these experiments, namely SafeRLHF and DiaSafety, to assess the safety of the responses generated. We employed PerspectiveAPI to automatically evaluate the safety of the responses. The API identifies five key aspects of safety probabilities: Toxicity, Insult, Profanity, Severe Toxicity, and Threat. In these categories, a lower score indicates better safety performance.
> For the LoRD model, we have retained the same hyper-parameters as those used in our domain-specific experiments to ensure consistency.
>
> **DiaSafety**:
>
>
> | Model               | Toxicity(%) | Insult(%) | Profanity(%) | Severe Toxicity(%) | Threat(%) |
> |---------------------|-------------|-----------|--------------|--------------------|-----------|
> | Llama3-8B (initial) | 14.20       | 7.94      | 8.35         | 1.58               | 2.29      |
> | Llama3-8B + MLE     | 8.31        | 3.69      | 4.31         | 0.83               | 1.50      |
> | Llama3-8B + LoRD    | **6.45**    | **2.81**  | **3.56**     | **0.71**           | **1.34**  |
>
>
>
> **SafeRLHF**:
>
>
> | Model            | Toxicity(%) | Insult(%) | Profanity(%) | Severe Toxicity(%) | Threat(%) |
> |------------------|-------------|-----------|--------------|--------------------|-----------|
> | Llama3-8B        | 7.92        | 2.71      | 2.80         | 0.30               | 1.49      |
> | Llama3-8B + MLE  | 4.87        | 1.98      | **1.66**     | **0.16**           | 1.02      |
> | Llama3-8B + LoRD | **3.55**    | **1.15**  | 2.84         | 0.38               | **0.79**  |

---

### Official Review · Reviewer_Xic1 · 2024-11-02

**Soundness:** 3
**Presentation:** 3
**Contribution:** 2
**Rating:** 5
**Confidence:** 3

**Summary:**

The paper discusses the vulnerabilities of large language models (LLMs) to model extraction attacks (MEAs). The authors propose a Locality Reinforced Distillation (LoRD) method via introducing the reinforcement learning procedures. LoRD costs less query times and mitigates watermark protection. Extensive experiments demonstrate the effectiveness of LoRD in extracting commercial LLMs. The paper also provides a theoretical analysis, discussing why LoRD can achieve stronger watermark resistance and higher query efficiency than existing methods.

**Strengths:**

1. The paper might be the first work to steal models by considering the alignment procedure of LLMs with RL.
2. Extensive and comprehensive experiments.
3. The paper provides theoretical analysis on the consistency of model performance.

**Weaknesses:**

Refer to the questions.
The experimental results are interesting. A more detailed analysis could be beneficial.

**Questions:**

1. Could you provide a more detailed analysis of Figure 6: Comparison of watermark resistance? According to Eq. 11, resistance across different \lambda values should be consistent. However, the results in Figure 6 show some inconsistencies in performance.

2. How does the choice of local model impact the final results? Since the goal of this paper is to steal the alignment capacity of a large commercial LLM, the capability of the foundational local model should be critical. Have you experimented with other models besides Llama3-8B?

3. In Table 1, for "Data to Text: E2E NLG Dušek et al. (2020) with 64 query samples," the stolen model (+LoRD) outperforms the victim model. Could this be due to overfitting? Could you analyze this further?

---

> ### Author Response · Authors · 2024-11-23
>
> Thanks for your review. We have revised our experiments part and provide feedback to address your concerns here:
>
> ### Abnormal Phenomenon on WMT (de-en) in Watermark Experiments
>
> We acknowledge that the watermark resistance experiment results on WMT (de-en) do not align perfectly with the setting of $\lambda_1$. We suspect that this abnormality is due to the fine-tuning checkpoints on the first two points, as the performance of LoRD on WMT (de-en) exhibits significantly higher variability compared to WMT (cs-en). This suggests that these two points may not have been properly trained. We will update the relevant experiments in due course. Nonetheless, even with this variability, LoRD still outperforms MLE in terms of watermark resistance on this dataset, so the conclusion of this part of experiments remains valid.
>
> ### Influence of Local Models
>
> **As in Appendix C.1.2 and Appendix C.1.3, we have already conducted experiments to investigate how different capacities of local models affect extraction performance.** Specifically, in Appendix C.1.2, we explored the relationship between the scale of local models and extraction performance using OPT series models with varying parameter numbers. In Appendix C.1.3, we evaluated the MEA efficacy across different local and victim models, including commonly used models such as Phi3, OPT, Qwen2, Mistral, and Llama3. Based on your feedback, we have strengthened this section of the experiments in the main paper.
>
> ### Explanation of Table 1's D2T Part
>
> We appreciate your detailed review of our paper and your questions regarding the "outperforms" situation in Table 1's D2T part. There are two reasons for it:
> 1. NLG evaluation is inherently challenging and may be subject to evaluation errors. As discussed in previous literature [1][2], current metrics like BLEU have limitations. Therefore, we used both lexical- and semantic-level metrics to provide a more comprehensive and convincing evaluation, as described in Section 5.1.
> 2. Different metrics may focus on different aspects of evaluation. Consequently, a bad answer may obtain a higher score in some metrics, but performs much worse on others. For example, if the local model generates a short sentence that is a subset of the reference sentence, it may receive an unreasonably high BLEU score. Such a abnormal phenomena can also be observed in BERTScore (Precision) and Rouge-L (Recall) for some extraction experiments.
>
>
> [1] A. R, P. Bhattacharyya, M. Sasikumar, and R. M. Shah, “Some issues
> in automatic evaluation of english-hindi mt: More blues for bleu,” 2006.
> [Online]. Available: https://api.semanticscholar.org/CorpusID:5690091
>
> [2] A. Stent, M. Marge, and M. Singhai, “Evaluating evaluation
> methods for generation in the presence of variation,” in Conference
> on Intelligent Text Processing and Computational Linguistics,
> 2005. [Online]. Available: https://api.semanticscholar.org/CorpusID:
> 11115098

---

> > ### Comment · Reviewer_Xic1 · 2024-11-27
> >
> > Thank you for providing additional explanations and experiments. The authors have addressed some of my concerns; however, question 1 remains somewhat unclear. On the other hand, the explanation for question 3 is clear and reasonable.

---

> > > ### Author Response · Authors · 2024-11-27
> > >
> > > Thank you for your valuable feedback.
> > >
> > > In response to Question 1, which concerns the abnormal results observed in the watermark experiments, we are currently re-training these experiments multiple times to ensure accuracy. Besides, we are preparing further watermark experiments to augment our findings. We expect to release the updated experimental results in a few days, along with a clearer and more persuasive analysis. Thank you!

---

> ### Author Response · Authors · 2024-11-28
> **A Further Explanation to Watermark Resistance Experiments**
>
> Regarding the watermark resistance experiments, we have retrained the local model with $\lambda\_1$ set to 0.0. Unfortunately, the experimental results continue to exhibit abnormalities, as illustrated in Figure 6. As elaborated in our paper, we suspect that this result arises from the disability of the regularization term when $\lambda\_1$ is set to zero, which concurrently explains the poor Rouge-L score observed in Figure 6. Consequently, for tasks necessitating the injection of substantial additional knowledge, utilizing $\lambda\_1$ diminishes the efficacy of the extraction process. From Figure 6, a reasonable range for setting $\lambda\_1$ appears to be between 0.2 and 0.6, with 0.5 serving as our default setting.
>
> To further investigate the correlation between watermark resistance and λ1​, we have conducted additional experiments on a different dataset (e2e-nlg), which exhibits a similar tendency to WMT (cs-en).
>
> | $\lambda\_1$ | P-value | Z-score | Rouge-L (F1) | BERTScore (F1) |
> |--------------|---------|---------|--------------|----------------|
> | 0.0          | 42.70   | 28.20   | 43.98        | 90.86          |
> | 0.2          | 39.86   | 39.22   | 44.08        | 90.84          |
> | 0.4          | 35.04   | 52.59   | 42.08        | 90.39          |
> | 0.6          | 38.35   | 43.55   | 43.42        | 90.79          |
> | 0.8          | 34.96   | 54.98   | 44.05        | 90.81          |

---

> > ### Comment · Reviewer_Xic1 · 2024-12-02
> >
> > Thanks for the continued experiments and responses. I find this paper interesting. However at this time, I would keep my score.

---

> > > ### Author Response · Authors · 2024-12-03
> > >
> > > Thanks for your response. We respect your feedback and opinion. Thanks!

---

### Author Response · Authors · 2024-11-23

Thank you to all the reviewers for your valuable feedback. We have thoroughly revised our work based on your comments and provided detailed, point-by-point responses to your questions.

Please do not hesitate to reach out if you have any further questions or concerns. Thanks!

For your reference, the original version of the submission is available for comparison at: https://anonymous.4open.science/r/LoRD-MEA-1EF2/v1.pdf

---

### Author Response · Authors · 2024-12-04
**Summary of the Review [1/2]**

We sincerely thank all reviewers for their valuable time and efforts during both the review and rebuttal periods. For the convenience of the reviewers' discussion and the Chairs' assessment, we provide a summary of the reviews below.

# Summary of the Weaknesses and the Concerns

We have made every effort to address the concerns raised by the four reviewers. Among them, two reviewers indicated that their concerns have been resolved, one reviewer highlighted one major concern that remains unresolved, and one reviewer did not provide any responses.

We have categorized these concerns into three main areas: (1) missing necessary experiments, (2) misunderstandings of the paper, and (3) requests for further discussion and additional experiments.

## Missing Necessary Experiments

- **Safety Alignment Extraction**: Reviewer Bcxr and Reviewer aNx7 suggested the need for a safety alignment extraction mechanism beyond domain-specific stealing. We agree with this necessity and have included alignment extraction experiments for two tasks in the revised version of the paper.

- **Ablation Study**: The reviewers expressed interest in an additional ablation study to validate the intuitions behind our loss function design. In response, we revised the methodology section of the paper and included an ablation study to address these concerns.

**Status**: Reviewer aNx7 acknowledged that these revisions well addressed their concerns, while Reviewer Bcxr has not provided feedback yet.

## Misunderstanding of the Paper

Several misunderstandings about the paper were identified, and we have sought to clarify them during the rebuttal:

1. **Impacts of the Choice of the Local Model**: We clarified that relevant experiments have already been presented in the Appendix.

2. **Limited Improvement to MLE**: We highlighted experimental results in Tables 1, 2, and 5, as well as Figures 7 and 8, to demonstrate that the improvements are substantial rather than limited.

3. **"Lack of Rigor and Explanation"** in the Design of Loss Functions and Theoretical Analysis: (i) We noted that many current works in the field of LLM+RL also lack rigor and explanation. In contrast, our study goes beyond intuition and empirical experiments by offering some in-depth theoretical explanations and analyses. (ii) We provided additional theoretical analysis in the revised version. (iii) We improved the readability of the paper based on prior studies.

**Status**: Point 1 was accepted by Reviewer Xic1, and Points 2 and 3(ii) were accepted by Reviewer 5Msi. The remaining points are still awaiting responses from Reviewer Bcxr.

## Discussion

1. **Abnormal Experimental Results**: Reviewer Xic1 raised concerns regarding abnormal points in the watermark resistance experiments (Figure 6) and the D2T experiments (Table 1). In response, we provided explanations and supplemented additional experiments to support our claims.
2. **Methodology Discussion**: Reviewer 5Msi proposed some interesting exploration of the methodology, particularly regarding the importance of the regularization term and the **rationale for not using direct prompt-based feedback**. We addressed these points by explaining our design choices and analyzing potential drawbacks of obtaining binary feedback using direct-intent prompts, which considers three core factors: stealthiness, query efficiency, and complexity.
We also supplemented two groups of experiments to support our analysis.
3. **Model-Level Watermarks**: We expanded the discussion in our paper on the potential utility of model-level watermarks for defending against our proposed method, addressing relevant concerns.

**Status**: Points 1 and 3 have been addressed. For Point 2, although our explanation with three reasons did not fully satisfy Reviewer 5Msi, the additional empirical attempt was provided as further clarification.


Additionally, the reviewers identified presentation issues in the paper, such as the incorrect placement of the "Ethical Statement" section. We appreciate their careful review and have revised the paper accordingly, which has been immensely helpful.

---

> ### Author Response · Authors · 2024-12-04
> **Summary of the Review [2/2]**
>
> # Summary of Strengths and Contributions
>
> We also highlight the positive feedback from reviewers regarding the strengths and contributions of the paper:
>
> - **Presentation and Organization**
>   - "... well-written with a clear structure and rich content, ... easy to follow." - Reviewer aNx7
> - "The paper is well-written, the tables and figures are well displayed. I thank the authors for their great efforts on well organizing their manuscript." - Reviewer 5Msi
> - **Contribution**
>   - "...might be the first work to steal models by considering the alignment procedure of LLMs with RL" - Reviewer Xic1
>   - "...seems promising and does try to fill in an important gap in the knowledge about model stealing attack against modern LLM" - Reviewer Bcxr
> - **Novelty and Impact**
>   - "an interesting problem with potential impact" - Reviewer Bcxr
>   - "The studied problem is practical and meaningful. The motivation is good and reasonable." - Reviewer 5Msi
>   - "...steal commercial models under a fully black-box threat model, making it highly practical" - Reviewer aNx7
>   - "...improves the way LLMs are extracted, thereby reducing the cost of queries." - Reviewer aNx7
>   - "...does not replicate possible watermarks in the victim model." - Reviewer aNx7
> - **Theoretical Analysis**
>   - "The paper provides theoretical analysis on the consistency of model performance." - Reviewer Xic1
> - **Experiments**
>   - "Extensive and comprehensive experiments..." - Reviewer Xic1
>   - "I like the analysis about Figure 5 (a spectrum of almost all NLP downstream tasks under stealing), which shows some novel and interesting insights." - Reviewer 5Msi
>   - "The Watermark Resistance part is interesting and reasonable." - Reviewer 5Msi
>   - "Although the method is not highly effective on every task, the authors have deeply explained the reasons behind these issues." - Reviewer aNx7
> - **Ethics Considerations**
>   - "...responsibly discussed ethical concerns and provided some possible defense strategies" - Reviewer aNx7
>
>
> # Summary of Scores
>
> | Reviewer | Soundness | Presentation | Contribution | Score (old) | Score (current) |
> |----------|-----------|--------------|--------------|-------------|-------|
> | Xic1     | 3         | 3            | 2            | 5           | 5     |
> | Bcxr     | 1         | 2            | 2            | 3           | 3     |
> | aNx7     | 3         | 3            | 3            | 6           | 6     |
> | 5Msi     | 2         | 4            | 2            | 3           | 5     |
>
>
> In summary, we sincerely thank all four reviewers again for their dedicated efforts and thoughtful feedback. We have carefully addressed all the questions and concerns raised by the reviewers, and based on the feedback from three of them, we confirm that we have resolved most of the issues. The reviews and rebuttal process truly enhanced the quality of the revised version of this paper. We kindly request that the reviewers reconsider their scores if our responses, explanations, and experiments have addressed your concerns. Thank you once again for your valuable input.

---

### Meta-Review · Area_Chair_cNdJ · 2024-12-11

**Metareview:**

This paper introduces an RL-based method called Locality Reinforced Distillation (LoRD) to reduce query complexity of LLM-targeted model extraction attacks. While the method itself is promising, reviewers identified major issues, including unclear presentation and insufficient experiments. Despite the authors' detailed rebuttal, the reviewers concluded that the current version does not meet ICLR's standards. I encourage the authors to continue refining their work for future submissions.

**Additional Comments On Reviewer Discussion:**

The reviewers remained silent throughout the rebuttal and AC-reviewer discussion phases. The AC carefully reviewed the paper, the comments from the reviewers, and the author rebuttal to form the final recommendation.

---

### Decision · Program_Chairs · 2025-01-22

Reject